



# IMK/IAA MIPAS temperature retrieval version 8: nominal measurements

Michael Kiefer[1], Thomas von Clarmann[1], Bernd Funke[2], Maya García-Comas[2], Norbert Glatthor[1], Udo Grabowski[1], Sylvia Kellmann[1], Anne Kleinert[1], Alexandra Laeng[1], Andrea Linden[1], Manuel López-Puertas[2], Daniel R. Marsh[3,4], and Gabriele P. Stiller[1]

[1]Karlsruhe Institute of Technology, Institute of Meteorology and Climate Research, Karlsruhe, Germany
[2]Instituto de Astrofísica de Andalucía, CSIC, Spain
[3]National Center for Atmospheric Research, Boulder, CO, USA
[4]Faculty of Engineering and Physical Sciences, University of Leeds, Leeds, UK

**Correspondence:** Michael Kiefer (michael.kiefer@kit.edu)

**Abstract.** A new global set of atmospheric temperature profiles is retrieved from recalibrated radiance spectra recorded with the Michelson Interferometer for Passive Atmospheric Sounding (MIPAS). Changes with respect to previous data versions include a new radiometric calibration considering the time-dependency of the detector non-linearity, and a more robust frequency calibration scheme. Temperature is retrieved using a smoothing constraint, while tangent altitude pointing information

is constrained using optimal estimation. ECMWF ERA-Interim is used as temperature a priori below 43 km. Above, a priori data is based on data from the Whole Atmosphere Community Climate Model Version 4 (WACCM4). Bias-corrected fields from specified dynamics runs, sampled at the MIPAS times and locations, are used, blended with ERA-Interim between 43 and 53 km. Horizontal variability of temperature is considered by scaling an a priori 3D temperature field in the orbit plane in a way that the horizontal structure is provided by the a priori while the vertical structure comes from the measurements. Additional

microwindows with better sensitivity at higher altitudes are used. The background continuum is jointly fitted with the target parameters up to 58 km altitude. The radiance offset correction is strongly regularized towards an empirically determined vertical offset profile. In order to avoid the propagation of uncertainties of $O_3$ and $H_2O$ a priori assumptions, the abundances of these species are retrieved jointly with temperature. The retrieval is based on HITRAN 2016 spectroscopic data, with a few amendments. Temperature-adjusted climatologies of vibrational populations of $CO_2$ states emitting in the 15 $\mu$m region are

used in the radiative transfer modelling in order to account for non-local thermodynamic equilibrium. Numerical integration in the radiative transfer model is now performed at higher accuracy. The random component of the temperature uncertainty typically varies between 0.4 and 0.8 K, with occasional excursions up to 1.3 K above 60 km altitude. The leading sources of the random component of the temperature error are measurement noise, gain calibration uncertainty, spectral shift, and uncertain $CO_2$ mixing ratios. The systematic error is caused by uncertainties in spectroscopic data and line shape uncertainties. It ranges

from 0.2 K at 24 km altitude for northern midlatitude nighttime conditions to 2.3 K at 12 km for tropical nighttime conditions. The estimated total uncertainty amounts to values between 0.5 K at 24 km and northern polar winter conditions to 2.3 K at 12 km and northern midlatitude day conditions. The vertical resolution varies around 3 km for altitudes below 50 km. The





long-term drift encountered in the previous temperature product has been largely reduced. The consistency between high spectral resolution results from 2002–2004 and the reduced spectral resolution results from 2005–2012 has been largely improved.

As expected, most pronounced temperature differences between version 8 and previous data versions are found in elevated stratopause situations. The fact that the phase of temperature waves seen by MIPAS is not locked to the wave phase found in ECMWF analyses demonstrates that our retrieval provides independent information and does not merely reproduce the prior information.

# 1 Introduction

The Michelson Interferometer for Passive Atmospheric Sounding (MIPAS) was a mid-infrared Fourier transform spectrometer operating in limb-viewing measurement geometry (Fischer et al., 2008). The spectral coverage was 4.15 to 14.6 $\mu$m (685–2410 cm$^{-1}$). From June 2002 to April 2012 spectrally resolved atmospheric emission spectra were measured globally from a polar sun-synchronous orbit, between 87.1° S and 89.3° N. The European Space Agency (ESA) has distributed a number of level-1b data versions which differ with respect to altitude registration, offset and gain calibration, nonlinearity correction,

spectral sampling and other issues relevant to the generation of calibrated geolocated radiance spectra. In this paper, we present vertical profiles of temperature retrieved from level-1b radiance spectra of ESA version 8.03 with the scientific level-2 MIPAS processor developed and operated by the Institute of Meteorology and Climate Research (IMK) at the Karlsruhe Institute of Technology, and the Instituto de Astrofísica de Andalucía (IAA). We shall use the terms "version 8" and/or "V8" to label this data version, and the same naming conventions apply mutatis mutandis to other data versions.

From 2002 to March 2004, MIPAS recorded interferograms with a maximum optical path difference (MOPD) of $\pm$ 20 cm, corresponding to a spectral resolution of 0.05 cm$^{-1}$ after Norton-strong apodization. This phase of operation will be called full spectral resolution (FR) period. Due to a technical defect, the MOPD had to be reduced to $\pm$ 8 cm after March 2004, leading to a spectral resolution of 0.125 cm$^{-1}$ (apodized). In turn, the vertical sampling has been increased. We shall call this second operation phase with a degraded spectral resolution reduced resolution (RR) period.

MIPAS measurements were taken in several measurement modes. The bulk of the data was taken with the nominal measurement mode (NOM), with an altitude coverage of approximately 6–70 km, and 17 and 27 tangent altitudes for FR and RR, respectively. Smaller parts of the data comprise a mode tailored for upper troposphere and lower stratosphere measurements (UTLS-1) with 19 tangent altitudes between 6 and 50 km. Further modes are designed for middle and upper atmosphere, and for noctilucent cloud measurements. Respective documentation of retrievals based on MIPAS level-1b data for these measurement

modes can be found in a companion paper (García-Comas et al., 2020).

A major change with respect to earlier level-1b data versions is the use of a model for the time dependency of the detector non-linearity. The response of the MIPAS detectors has shown to become more linear with time. However, previous calibration relied solely on a preflight detector characterization. Related investigations are summarized in Section 2.

The spectrally resolved radiance measurements are converted into global temperature distributions. The precision and accu-

racy of the resulting data depends largely on the adequateness of the setup of the retrieval. Several improvements have been



made with respect to this, which are discussed in Section 3. The uncertainty budget is presented in Section 4. Exemplary results are shown in Section 5, and in Section 6 we summarize our findings, critically discuss the success of the attempt to improve the MIPAS temperature retrieval, identify unsolved problems, and suggest topics for future work.

We intentionally describe the retrieval on a rather technical level in order to help the users to better understand the data and in order to provide a basis and documentation for scientists building future retrieval processors.

## 2   Change in level-1b gain calibration

The long-wavelength detectors A1, A2, B1 and B2, which form the MIPAS bands A, AB, and B, are affected by non-linearity, whereas the short-wavelength bands C and D have linear detectors. Preflight measurements of detector non-linearity have been available and were used for calibration of all data versions up to version 6. MIPAS temperature time series of these data ver-
sions have shown drifts which have been analyzed by Eckert (2012); Penckwitt et al. (2015); McLandress et al. (2015); Laeng et al. (2020). These drifts are attributed to an inadequate non-linearity correction. Birk and Wagner (2010) found that detector nonlinearity is a function of time, i.e. that the detector response became more linear over the MIPAS lifetime. These authors have provided an age-dependent correction scheme which has been applied since data version 7. Its adequacy has been proven by the fact that it removes the major fraction of the drifts found in previous data versions. For ozone, it has been shown that
application of this new correction scheme improves the long-term stability (Laeng et al., 2018). However, temperature validation has revealed that the version 7 non-linearity correction led to too small values, especially at the beginning of the mission (Hubert et al., 2016). Therefore the non-linearity correction has been reviewed again, and the estimate for the modulation efficiency has been improved (Kleinert et al., 2015). This leads to generally higher radiances and a better agreement with the validation data.

A further change in the calibration procedure refers to the selection of calibration spectra. While up to version 7 the gain calibration was updated only once per week, in version 8 all available gain calibration measurements were used, which is usually one gain measurement per day. Thus, most version 8 MIPAS radiance spectra rely on calibration spectra from the same day. In the case that MIPAS operation was interrupted, care was taken that the gain measurements closest in time from the same instrument state were applied.

## 80  3   Retrieval

In this paper, the most recent temperature data versions generated with the MIPAS processor developed and operated at the Institute of Meteorology and Climate Research (IMK) in cooperation with the Instituto de Astrofísica de Andalucía (IAA) are discussed. The IMK-IAA data processor relies on multi-parameter non-linear least squares fitting of measured and modeled spectra (von Clarmann et al., 2003a). Its extension to retrievals involving non-local thermodynamic equilibrium emissions is
described in Funke et al. (2001), while the adaption to the RR measurements is documented in von Clarmann et al. (2009). In the following we discuss all retrieval settings relevant for temperature, which is retrieved along with the tangent altitude of the





line of sight. The combined temperature and tangent altitude retrieval is the first step in the chain of retrievals and is preceded only by the determination of a frequency shift (see Section 3.2).

## 3.1 Retrieval Method

In order to put the improvements discussed later into the context of the pre-existing retrieval scheme, we recapitulate the main features of the MIPAS temperature retrieval scheme here. MIPAS spectra are analyzed with constrained nonlinear least squares fit. The updated guess of the state vector $\boldsymbol{x}_{i+1}$ at iteration $i+1$ is calculated from the previous estimate $\boldsymbol{x}_i$ as

$$
\begin{aligned}
\boldsymbol{x}_{i+1} \quad = \quad & \boldsymbol{x}_i + \left( \mathbf{K}^T \mathbf{S}_{\text{y,noise}}^{-1} \mathbf{K} + \mathbf{R} \right)^{-1} \\
& \left( \mathbf{K}^T \mathbf{S}_{\text{y,noise}}^{-1} \left( \boldsymbol{y} - \boldsymbol{F}(\boldsymbol{x}_i) \right) - \mathbf{R} \left( \boldsymbol{x}_i - \boldsymbol{x}_a \right) \right).
\end{aligned}
\tag{1}
$$

Here $\mathbf{K}$ is the Jacobian containing the partial derivatives $\frac{\partial y_m}{\partial x_n}$; superscript $T$ indicates transposed matrices; $\mathbf{S}_{\text{y,noise}}$ is the covariance matrix representing measurement noise; $\mathbf{R}$ is a regularization matrix; $\boldsymbol{y}$ is the vector of measurements under consideration; $\boldsymbol{F}(\boldsymbol{x}_i)$ is the vector of the respective simulated measurements, based on the Karlsruhe Optimized and Precise Radiative Transfer Algorithm (KOPRA, Stiller, 2000); and $\boldsymbol{x}_a$ is the vector of prior information on $\boldsymbol{x}$. The vertical grid for the retrieval of the temperature profile is 0, 4[1]50, 52[2]70, 72.5[2.5]80, 85[5]110, 120 km (notation: lower altitude [altitude step] upper altitude). MIPAS measurements are analyzed limb profile by limb profile in a global-fit mode in a sense that all fitting residuals related to an entire limb scan are minimized simultaneously rather than in sequence (Carlotti, 1988). Contrary to the original global-fit approach, which was an un-regularized maximum likelihood retrieval, we do use regularization (Section 3.3). Adequate a priori information above the uppermost MIPAS tangent altitude proved to be of particular importance (Section 3.4). Also contrary to the original global-fit method, our retrieval scheme supports consideration of horizontal variability along the line-of-sight direction (Section 3.5), where the respective element of $\boldsymbol{x}$ associated with a certain altitude is a scaling factor for the horizontal temperature distribution. Temperature is fitted jointly with a correction of the tangent altitude of the line of sight in order to minimize mutual error propagation. The temperature and tangent altitude fit uses spectral lines of $CO_2$, because excellent prior knowledge on the vertical distribution of this gas is available, and because no rapid changes of its mixing ratios are expected. The retrieval relies on specific parts of the spectra, called "microwindows", which contain maximum information on the target quantities, but are least interfered of gases of unknown abundancy (see Section 3.6). This means that $\boldsymbol{y}$ contains only a subset of the available spectral measurements. However, $CO_2$ lines are not perfectly isolated but interfered by a background continuum and some signal of mainly $H_2O$ and $O_3$. To avoid propagation of uncertainties of the a priori uncertainties of these gases and the background continuum, these parameters are jointly fitted with the temperature and tangent altitude information, i.e., they are part of the $\boldsymbol{x}$ vector (Sections 3.7 and 3.9, respectively). Also an additive offset calibration correction is retrieved, which is frequency-independent for each microwindow (Section 3.8). An accurate retrieval depends crucially on the choice of reliable spectroscopic data (Section 3.10). Consideration of non-local thermodynamic equilibrium emission improves the retrieval at higher altitudes (Section 3.11). Adequately chosen numerical accuracy parameters are important for a sound trade-off between the accuracy of the results and computational efficiency (Section 3.12).





Previous MIPAS temperature retrievals were described in von Clarmann et al. (2003a) for FR measurements and in von

Clarmann et al. (2009) for RR nominal mode version 4 measurements. The RR upper troposphere/lower stratosphere measurements mode version 4 retrievals were documented by Chauhan et al. (2009). RR version 5 nominal mode retrievals were presented by von Clarmann et al. (2013). Version 3 temperatures were validated by Wang et al. (2005, 2004), while version 5 temperatures were validated by Stiller et al. (2012).

### 3.2 Frequency calibration

Prior to the retrieval of atmospheric state variables, a frequency shift scale is determined from the spectra. In principle, the level-1b data is already frequency calibrated, but frequency calibration and instrument line shape modelling are intertwined. For our retrieval we need an adjustment of the frequencies which accounts also for any frequency shift implied by the modelling of the instrument line shape.

A technical change with respect to the frequency calibration has been implemented in the level-1b processing. Instead of one

spectral calibration per four scans, spectral calibration in version 8 is performed once per day, relying on measurements of one full orbit. The mean spectral shift scale is then applied to all measurements of the respective day. The MIPAS spectral accuracy of ESA's frequency calibration is estimated to $0.00065 \, \text{cm}^{-1}$ at $2410 \, \text{cm}^{-1}$ (Kleinert et al., 2018). The described modification of the frequency calibration scheme leads to very small differences in the retrieved atmospheric state variables and is of minor relevance to the IMK/IAA MIPAS processing, because a spectral shift correction is performed anyway as the first step of the

retrieval chain.

The calculation of the spectral shift within the IMK-IAA processing is made by minimizing the residual between simulated and measured spectra using Eq. (1) and then fitting a linear regression function to the shift values, which are calculated for the single microwindows. The microwindows used for the frequency shift retrieval are shown in Table 1. Line selection criteria were sufficient line strength, good separation of the target lines, and a good coverage of the MIPAS A, AB, B and C

bands. The MIPAS D band was not used because sufficiently strong D-band transition would require consideration of non-local thermodynamic equilibrium in the respective radiative transfer calculations, which was considered as too costly for this purpose.

For the frequency scale correction, we use spectra at a tangent altitude of 38 km, where the lines are narrow enough to allow for a good spectral calibration and where the signal is still strong enough to avoid large random error in the spectral

shift correction. Under some polar winter conditions, and particularly in the case of an elevated stratopause, however, the shift retrieval from a single spectrum is by far not precise enough. In order to reduce large fluctuations due to noise, the spectral shift retrieval is constrained towards its temporal mean, where separate means were used for the high-resolution measurement period and the reduced-resolution measurement period. The reason for this choice was the use of different instrument line shapes for these measurements.

Inclusion of $NO_2$ proved to be essential, particularly because of a sizeable signal of mesospheric $NO_2$. Omission of this gas gas would lead to artificial diurnal variations of the spectral shift.





**Table 1.** Microwindows for the retrieval of residual spectral shift.

| lower boundary $(\text{cm}^{-1})$ | upper boundary $(\text{cm}^{-1})$ | target line |
|---|---|---|
| 690.0000 | 695.0000 | $CO_2$ |
| 801.0000 | 805.0000 | $CO_2$, $O_3$ |
| 940.0000 | 945.0000 | $CO_2$ |
| 1076.0000 | 1080.0000 | $O_3$, $CO_2$ |
| 1145.0000 | 1148.0000 | $O_3$ |
| 1240.0000 | 1250.0000 | $CH_4$, $H_2O$ |
| 1338.0000 | 1340.0000 | $H_2O$, $CH_4$ |
| 1488.0000 | 1490.0000 | $H_2O$ |
| 1589.0000 | 1597.0000 | $H_2O$, $NO_2$ |
| 1746.0000 | 1748.0625 | $H_2O$ |

The fit is carried out using a maximum a posteriori scheme Rodgers (2000) using the mean of the spectral calibration scale over the entire FR and RR period, determined in a previous step, as a priori, and a priori variances of $(0.00035 \text{ cm}^{-1})^2$ for the FR and $(0.0007 \text{ cm}^{-1})^2$ for the RR measurements). The actual spectral correction for any wavenumber can be determined using the linear regression function.

### 3.3 Regularization

According to the retrieval vector $x$, $\mathbf{R}$ has a block-diagonal structure, and the choice of the regularization can be made independently for each group of variables.

In general we use a regularization term which is composed of a smoothing component $\mathbf{R}_{\text{smooth}}$ and a diagonal component $\mathbf{R}_{\text{diag}}$:

$$\mathbf{R} = \mathbf{R}_{\text{smooth}} + \mathbf{R}_{\text{diag}}. \tag{2}$$

Here the diagonal component $\mathbf{R}_{\text{diag}}$ is formally equivalent to the inverse of an a priori covariance matrix without information on inter-altitude correlations. For the regularization term $\mathbf{R}_{\text{smooth}}$ following implementation of the altitude dependence

$$\mathbf{R}_{\text{smooth}} = \mathbf{L}^T \begin{pmatrix} \gamma_1 & 0 & \dots & 0 \\ 0 & \ddots & & 0 \\ \vdots & & \ddots & \vdots \\ 0 & \dots & 0 & \gamma_{N-1} \end{pmatrix} \mathbf{L} \tag{3}$$





has replaced the approach by Steck and von Clarmann (2001), which has been used in our retrievals up to version 5. Here $\mathbf{L}$ is an $(N-1) \times N$ first order finite difference operator as suggested by Tikhonov (1963); Twomey (1963); Phillips (1962), but scaled with the respective gridwidth to yield difference quotients. The $\gamma$-values control the altitude dependence of the strength of the regularization.

    The regularization term used for the parameter temperature is $\mathbf{R}_T = \mathbf{R}_{\mathrm{smooth}}$ with the values of all $\gamma_i$ being set to 0.49 K$^{-2}$
in the entire altitude range.

    The tangent altitudes are constrained towards the line-of-sight engineering information. The respective block of $\mathbf{R}$ can be understood as an inverse a priori uncertainty covariance matrix describing both the relative pointing uncertainties between adjacent tangent altitudes and the absolute pointing uncertainty of the entire limb scan as a whole. The relative pointing a priori uncertainties were assumed to be 60 m in the RR measurement mode and 150 m in the FR measurement mode, in terms of $1\sigma$
standard deviations. The standard deviation of the absolute pointing uncertainty, representing the possible altitude shift of the entire limb scan, was assumed to be 900 m.

### 3.4   A priori temperature and trace gas distributions

In older nominal mode retrieval versions problems occurred which could be traced back to the use of inadequate a priori temperature distributions for altitudes above the uppermost MIPAS tangent altitude. Here, neither reliable analysis data are
available, nor can MIPAS vertically resolve the temperature profile. Older retrieval versions used the NRLMSISE-00 climatology (Picone et al., 2002) at these altitudes. However, this climatology has systematic biases (Emmert et al., 2020) and does not capture short-term variations occuring in dynamically active episodes such as elevated stratopause events. Due to missing MIPAS measurement information at related altitudes, this error propagated into the MIPAS temperatures in the nominal scan range. These temperature retrieval errors further propagated noticeably into retrievals of trace species, e.g. ozone (Laeng et al.,
185   2018).

    For IMK/IAA MIPAS version 8 temperature retrievals ECMWF ERA-Interim analysis fields (Dee et al., 2011) were used as a priori at altitudes up to 43 km, because ERA-5 was not available for the MIPAS time period when the processing was started. A priori temperatures above 53 km are based on Whole Atmosphere Community Climate Model (WACCM, Marsh, 2011; Marsh et al., 2013) Version 4 (WACCM4) fields of a specified dynamics run (García et al., 2017), which provided
output specifically for the MIPAS measurement geolocations and times. Since a specified dynamics run was used, the actual atmospheric conditions including stratospheric warming events and elevated stratopauses were sufficintly well reproduced. The WACCM temperatures were bias-corrected using MIPAS version 5 middle and upper atmosphere measurements, which cover an altitude range of 18–102 km, but are performed less frequently (García-Comas et al., 2014). Multi-annual averages of MIPAS-collocated WACCM differences were used to construct an altitude- and latitude-dependent seasonal correction,
independently for am and pm observations. Between 43 and 53 km, a smooth transition between ECMWF and bias-corrected WACCM temperatures is obtained by linear interpolation along with hydrostatic correction of pressures at the given geometric altitudes.





$CO_2$ distributions are imported from an SD-WACCM4-based climatology. From MIPAS V5 data gas profiles are generated for interfering species, and for initial guess profiles for $O_3$ and $H_2O$, which both are jointly fitted together with target state

variables. The a priori of the latter is a zero profile, while the regularisation is of Tikhonov type.

### 3.5 Horizontal variability

Typically, a locally spherically symmetric atmosphere is assumed in profile retrievals. That is to say, within one profile retrieval the atmospheric state is assumed to be a function of altitude only and does not vary with latitude or longitude. Since limb measurements used for one profile retrieval cover, depending on the measurement mode, about 1600 to 2200 km in the horizontal,

this horizontal homogeneity assumption is not without problems. Depending on the computational effort spent on accurate radiative transfer modelling, a fully tomographic retrieval as suggested by Carlotti et al. (2001, 2006) or Steck et al. (2005) often is beyond reach. As a first step, horizontal inhomogeneities of temperature have been considered in the trace gas retrievals since MIPAS version 4 by retrieving a horizontal temperature gradient applicable in a range of $\pm 400$ km around the tangent point (Kiefer et al., 2010). For retrievals based on level-1b spectra of version 7 onwards we go a step further and consider a

full a priori 3D temperature field, generated from ECMWF ERA-Interim data, extended by NRLMSISE-00 data above 60 km. During the retrieval, the temperatures of this 3D a priori field are scaled at each altitude in a way that the horizontal structure is provided by the a priori while the vertical structure comes from the measurements. The respective component of the retrieval vector $x$ is the 1D vector of scaling factors. Roughly speaking, the result is a temperature profile which provides the best spectral fit under the assumption that the a priori horizontal structure of the temperature field is correct. The information on the

horizontal temperature variability enters through the a priori but the vertical structure is provided by the MIPAS temperature retrievals.

Additionally, the retrieval of a horizontal gradient directly from the spectra of a single limb sequence is performed. However, the horizontal gradients are strongly regularized towards zero below 60 km, where ECMWF ERA-Interim temperature fields are available, and above 70 km, the topmost tangent altitude of MIPAS nominal measurement mode. In between, the regularization

of the temperature gradient is chosen weaker in order to better exploit the information on the horizontal temperature gradient provided by the measurements.

### 3.6 Microwindows

The retrieval does not use the entire measurement data but only parts of the spectra which are particularly sensitive to the target species, so-called 'microwindows' (see, von Clarmann and Echle, 1998 for the rationale behind this approach). For the

combined temperature and tangent altitude retrieval $CO_2$ lines are used, because the mixing ratio distribution of $CO_2$ is well known and only weakly structured. This reduces the number of unknowns in the retrieval.

In order to have more information on temperature at high altitudes, additional microwindows were included since data version V5. These are 686.8125–689.7500 cm$^{-1}$, 689.8750–692.6250 cm$^{-1}$, 699.4375–702.3750 cm$^{-1}$, 719.6250–722.5000 cm$^{-1}$, 740.3750–742.8750 cm$^{-1}$, and 791.1875–792.6875 cm$^{-1}$ (given for the wavenumber grid of the RR measurements).

The full list of microwindows for FR and RR measurements is presented in Table 2. The inclusion of $CO_2$ Q-branches on





**Table 2.** Microwindows of the combined MIPAS temperature and tangent altitude retrieval for the full (first column) and reduced (second column) spectral resolution.

| Wavenumber range (FR) ($\mathrm{cm^{-1}}$) | Wavenumber range (RR) ($\mathrm{cm^{-1}}$) | Altitude Range (km) |
|---|---|---|
| 686.9500–689.0500 | 686.8125–689.7500 | 42–100 |
| 690.1250–692.2500 | 689.8750–692.6250 | 42–100 |
| 699.8750–701.8750 | 699.4375–702.3750 | 42–100 |
| 719.6250–721.0500 | 719.6250–722.5000 | 33–100 |
| 731.2500–731.8000 | 731.2500–731.8125 | 21–72 |
| 741.2000–741.8000 | 740.3750–742.8750 | 33–69 |
| 744.3250–745.5000 | 744.3125–745.5000 | 21–72 |
| 749.5000–749.8000 | 748.9375–749.8125 | 18–72 |
| 765.8750–766.5500 | 765.8750–766.5625 | 21–72 |
| 780.4500–780.6250 | 780.4375–780.6250 | 6–72 |
| 791.2000–791.8750 | 791.1875–792.6875 | 18–63 |
| 798.1250–798.5000 | 798.1250–798.5000 | 21–72 |
| 810.8250–811.0500 | 810.8125–811.0625 | 6–72 |
| 812.2500–812.5500 | 812.2500–812.5625 | 6–72 |

the one hand implies the consideration of line mixing (omitted in previous data versions), but on the other hand allows to use the same microwindow selection for analysis of MIPAS nominal and middle atmosphere measurements (García-Comas et al., 2020). So apart from the increased information gain for higher altitudes, this choice will lead to a better inter-consistency between the two datasets. Depending on the tangent altitude, certain data points within a microwindow can be discarded to
avoid interference by other than $CO_2$ lines.

### 3.7    The background continuum

Joint-fitting of a background radiance continuum has always been a standard feature of all MIPAS retrievals (e.g., von Clarmann et al., 2003a). The purpose of this is to account for all radiance contributions which are neither included in the line-by-line calculation of absorption cross-sections nor by the pressure-temperature interpolation of pre-tabulated laboratory measurements
of absorption cross-sections of heavy molecules. These contributions are caused by (a) the far-wing contributions of transitions spectrally distant from the analysis window under investigation which add up to a continuum-like background signal whose line-by-line modelling would be inefficient; (b) continuum emission of trace gases by, e.g., dimers in the case of $H_2O$; (c) differences between the idealized modeled line-shapes and the true super- or sub-Lorentzian pressure broadening; and (d) the emission by non-gaseous components of the atmosphere like clouds, aerosols, volcanic ash or meteoric dust. Since these non-



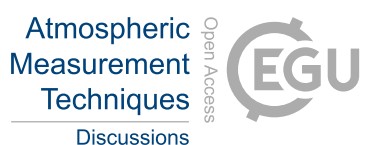

line-by-line effects are mostly important in the lower atmosphere, the background continuum contribution was only fitted up to 33 km altitude in previous data versions and set to zero above. It turned out, however, that consideration of the background continuum up to altitudes of 58 km significantly improved the robustness of the retrievals and removed known biases in retrieved state variables. The cause of the continuum signal from high altitudes is presumably meteoric dust (Neely III et al., 2011). The relevance of a high-reaching continuum signal was first discovered by Haenel et al. (2015) in the context of the

retrieval of $SF_6$.

Only a smoothing constraint is applied to the continuum retrieval up to 58 km, without any diagonal term. Above, the continuum is regularized exclusively by a diagonal term and an apriori of zero. Formally, an individual continuum-profile is retrieved per microwindow, but the continuum values are not only constrained in the altitude domain but also in the frequency domain. The latter smoothing constraint avoids unrealistic jumps of the value of the background continuum between adjacent

microwindows.

### 3.8  Offset correction

Besides the background continuum, we retrieve also a radiance offset profile which is meant to correct the radiance zero level calibration. While the continuum is additive to the absorption coefficient and appears in the exponent of Beer's law, the offset correction is directly additive in the radiance space. When radiative transfer is linear, which is the case for high tangent altitudes,

the offset correction and the background continuum cannot be distinguished and the simultaneous retrieval of both leads to a nullspace of solutions. This problem is solved by strongly constraining the background continuum to zero above 58 km, while the vertical offset profile is strongly regularized towards an empirically determined offset correction profile (Kleinert et al., 2018), which is used as a priori for the fit of the zero level correction. The actual offset per microwindow and per altitude is retrieved using both $\mathbf{R}_{\mathrm{smooth.}}$ and $\mathbf{R}_{\mathrm{diag.}}$. The diagonal term corresponds to a variance roughly a factor of two larger than the

offset uncertainty obtained by Kleinert et al. (2018), in order to account for possible unknown uncertainties. No regularization of the offset in the frequency domain has been applied, i.e., the offset can vary independently between microwindows.

### 3.9  Joint fit of $O_3$ and $H_2O$

Ideally, microwindows contain only signal of the target species and are free of any interfering signal. In general, however, such microwindows do not exist. In particular, $H_2O$ and $O_3$ have sizeable contributions in the microwindows of the temperature

retrieval. Since the temperature retrieval is the first step in the retrieval chain, no actual information on the highly variable trace gas abundances is available.

To avoid the mapping of inadequate assumptions on the actual $H_2O$ and $O_3$ abundances, these species' mixing ratio profiles are jointly retrieved with temperature. Since the microwindows of the temperature retrieval have not been optimized for joint retrieval of $H_2O$ and $O_3$, the resulting mixing ratios are discarded. The only purpose of this joint-fit approach is to avoid related

error propagation.





### 3.10  Spectroscopy

The HITRAN 2016 spectroscopic database (Gordon et al., 2017) was used for $CO_2$, whose lines provide the information on temperature and the tangent altitude, as well as for most interfering species. Exceptions are $O_3$ and $HNO_3$, for which the dedicated MIPAS spectroscopic database, provided by Flaud et al. (2003) was used.

### 280  3.11  Non-local thermodynamic equilibrium

Typically, radiative transfer in the stratosphere is calculated assuming that the atmosphere is in local thermodynamic equilibrium (LTE). Test calculations, however, have provided evidence that the consideration of non-LTE (NLTE) populations of vibrational states involved in the contributing $CO_2$ bands makes a difference also for temperature retrievals in the MIPAS nominal observational altitude range. The non-LTE effects are only moderate here and thus a full-blown non-LTE retrieval using all the machinery developed by Funke et al. (2005) seems undue. Instead we use a non-LTE parameterization that accounts for the temperature dependence of vibrational non-LTE populations in an approximate manner (manuscript in preparation) which is briefly explained in the following.

Considering a simple 2-level system under non-LTE conditions, upper and ground state populations $n_1$ and $n_0$, respectively, are related by

$$n_1 = \frac{P+R}{L+A}\, n_0 \tag{4}$$

with the collisional productions and losses $P$ and $L$, respectively, radiative losses $A$, and non-thermal productions $R$ (e.g., radiative production by solar absorption). In this equation, only $P = L\exp(-\Delta E/kT)$, with $\Delta E$ being the energy difference between upper and ground state, is temperature dependent. Hence, Eq. 4 can be separated in a temperature dependent term $a\exp(-\Delta E/kT)$ and a temperature independent term $b$ with $a = L/(L+A)n_0$ and $b = R/(L+A)n_0$.

The radiative transfer algorithm uses population ratios $r = n_{\mathrm{NLTE}}/n_{\mathrm{LTE}}$ with $n_{\mathrm{LTE}} = n_0\exp(-\Delta E/kT)$. Using Eq. 4 and the identity $b\exp(\Delta E/kT) = r - a$, the population ratio $r(T,z)$ can be expressed as function of the ratio $r(T_0,z)$ at a reference temperature $T_0$ as

$$r(T,z) = U(z) + (r(T_0,z) - U(z))\exp\left[\frac{E(z)}{k}\left(\frac{1}{T(z)} - \frac{1}{T_0(z)}\right)\right] \tag{5}$$

with $U = a$ and $E = \Delta E$ for the simple case described above. An updated version of the Generic RAdiative traNsfer AnD non-LTE population algorithm (GRANADA) (Funke et al., 2012) computes the parameter profiles $U(z)$ and $E(z)$ for realistic and more complex situations (i.e., multi-level systems, non-linear interactions by VV collisions, etc.), allowing for a temperature parameterization of non-LTE population ratios in a local approximation. A seasonal and latitudinal climatology of $r(T_0,z)$, $U(z)$ and $E(z)$ for the local times of MIPAS ascending and descending overpasses has been calculated offline with GRANADA and is considered by the KOPRA radiative transfer model in the forward calculations (Funke and Höpfner, 2000) to estimate the non-LTE population ratios of vibrational states 01101, 02201, 10011, and 11101, involved in the observed $^{16}C^{12}O_2$ bands for the actual temperatures, at each line-of-sight path segment during the retrieval iterations. This approach seems to be a fair compromise between rigor and efficiency.





### 3.12 Numerical issues

The accuracy of the numerical integration in the radiative transfer modelling has been improved in several places. In order to achieve a more accurate numerical integration of the radiance over the field of view, now 5 pencil beams are use throughout, while older retrievals (up to version 5) used only three pencil beams at some tangent altitudes. Also, in order to improve the numerical accuracy, a finer wavenumber grid is used for calculation of the monochromatic absorption cross sections ($0.00048828125$ cm$^{-1}$ instead of $0.001$ cm$^{-1}$). The convolution of the spectrum with the apodization function (Norton and Beer, 1976) now includes a wider wavenumber range. Additionally, a more conservative rejection threshold for lines so small that they are deemed not to contribute in any sizeable way to the total signal has been chosen. Further, it was found that it is advantageous to recalculate the absorption cross-sections during each iteration in the first seven layers above each tangent altitude. Formerly this costly line-by-line calculation was performed only during the first iteration and the cross-sections were re-used in all layers except the layer above the tangent altitude of the respective line of sight. However, when the temperature profile varies from iteration to iteration, the mass-weighted mean temperatures and pressures of the respective layer change, which is better accounted for by the new approach.

In the retrieval code an 'oscillation detection' has been activated which identifies failure of convergence in the sense that the iteration flips back and forth between two minima of the cost function according to $x_{i+1} \approx x_{i-1}$ and $x_i \approx x_{i-2}$. In this case $x_{i+1}$ is set to $\frac{x_{i+1}+x_i}{2}$, and one further iteration step is performed.

In version 8, 99.95% of the retrieval converged successfully. This is an improvement compared to versions 5 and 7, with 99.37% and 99.85% convergence rate, respectively.

## 4 Error budget

The error budget of MIPAS temperatures for nine atmospheric conditions is illustrated in Figs. 1–2 and reported in the Appendix in Tables A1–A9. The atmospheric conditions under consideration are northern and southern polar winter, polar summer, northern and southern midlatitudes for day and night, and tropical day and night.

The relevant sources of error are measurement noise, gain calibration, frequency calibration (spectral shift), mixing ratios of $CO_2$, uncertainties in spectroscopic data and the spectral line shape of the instrument. We first discuss the relevant error sources of the MIPAS temperature retrieval and report the input of assumed uncertainties of the error estimation. In order to comply with the TUNER (Towards Unified Error Reporting, von Clarmann et al., 2020) recommendations, we report uncertainties of chiefly random nature and systematic nature separately (Sections 4.2 and 4.3, respectively). All reported uncertainties are standard deviations ($1\sigma$).

Every single profile retrieval comes with a noise estimate, while parameter errors, model errors and so forth are provided as mean uncertainties for the representative conditions listed above. The total estimated error ranges from 0.5 K at 24 km and northern polar winter conditions to 2.3 K at 12 km and northern midlatitude daytime conditions.

In general the uncertainties are small in the lower stratosphere and then slowly increase towards higher altitudes (see Figs. 1–2). They also increase towards the tropopause region, and exhibit a strong increase below. This explains, together with the





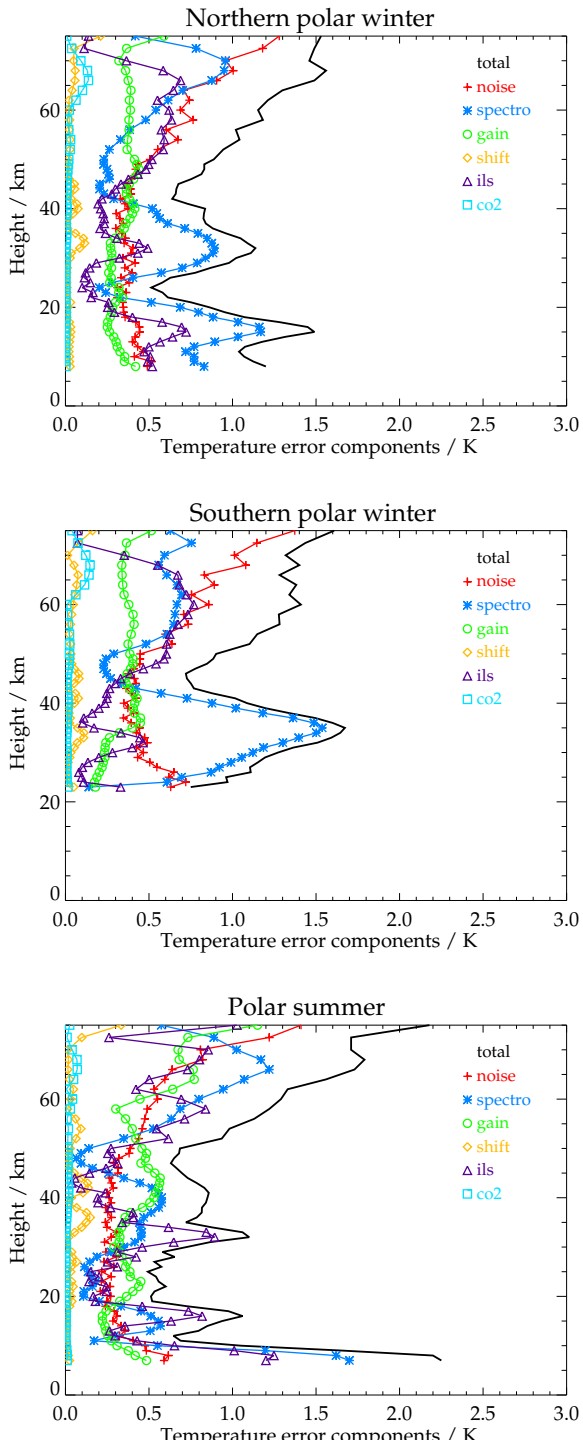

**Figure 1.** Temperature error budget for northern polar winter (top panel), southern polar winter (middle panel) and polar summer (bottom panel) atmospheres. All error estimates are 1-$\sigma$ uncertainties. Error contributions are marked spectro for spectroscopic error, gain for gain calibration error, shift for spectral shift error, ils for intrument line shape error, and co2 for error due to $CO_2$-VMR uncertainty.





**Figure 2.** Temperature error budget for northern midlatitudes daytime (top left panel), southern midlatitudes daytime (top right panel), northern midlatitudes nighttime (middle left panel), southern midlatitudes nighttime (middle right panel), tropical daytime (lower left panel), and tropical nighttime (lower right panel) atmospheres. All error estimates are 1-$\sigma$ uncertainties.





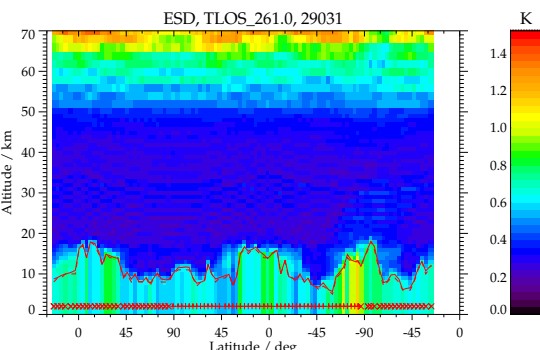

**Figure 3.** The noise-component of temperature error in Kelvin for sample orbit 29031 as a function of latitude and altitude. The red line is the lower edge of the instrumental field of view at the lowermost tangent altitude used. Temperatures retrieved below this altitude are solely determined by the retrieval constraint.

variation of the tropopause altitude, why errors for a given altitude in the tropopause region were found to vary largely between different limb scans. The retrieval proves to be particularly susceptible to errors just above the lowermost tangent altitude used. This illuminates why error estimates for northern and southern nighttime midlatitudes (middle panels in Fig. 2) differ so much in the tropopause region. This difference is merely caused by the different fraction of useful (not cloud-contaminated) limb
345 scans reaching down into the troposphere. This behaviour is also seen in Figure 3 which shows the propagation of measurement noise into the retrieved temperatures. High uncertainties are found just above the lowermost tangent altitude used (red solid line). The implication for the data user is that error estimates in the tropopause region can be regarded as fairly reliable in a statistical sense but can deviate for single profiles as described above.

## 4.1 Error sources

350 Following the terminology of von Clarmann et al. (2020) we distinguish measurement errors, parameter errors, and model errors. Measurement errors include measurement noise and all uncertainties related to less than perfect knowledge of the instrument state. Parameter errors are uncertainties of atmospheric state parameters which are assumed to be sufficiently well known and thus not treated as unknowns of the retrieval. Model errors include deficiencies in the way the radiative transfer model describes radiative transfer through the atmosphere and uncertainties in constants such as spectroscopic data. We do
355 not evaluate the smoothing error because we conceive the retrieval as an estimate of the smoothed true state rather than a smoothed estimate of the true state (see, Rodgers, 2000, Section 3.2.1, for a discussion of this issue). Furthermore, we provide information on error correlations in various domains (Section 4.4) and averaging kernels (Section 5.1).





### 4.1.1 Measurement errors

The following measurement errors were found to make a sizeable contribution to the overall error budget: Measurement noise,
gain calibration error, instrument line shape uncertainty, and frequency calibration (spectral shift) uncertainties. Error propagation was performed using linear theory, applied to forward radiative transfer. The propagation of measurement noise was evaluated by means of Eq. 20 of von Clarmann et al. (2020), while the propagation of other measurement errors was estimated on the basis of sensitivity studies for the given atmospheric conditions.

Measurement noise, as estimated from the imaginary part of the spectra, is reported in the level-1b data. In the spectral region used for the temperature retrievals, values are in the range 15–33 nW/(cm$^2$ sr cm$^{-1}$) after apodization.

Gain uncertainties were estimated from scaling ratios between overlapping channels deduced from dedicated IF16 measurements over the mission (see Fig. 11 of Kleinert et al. (2018)). They are estimated to be 1.4% during the FR period and 1.1% during the RR period. For the instrument line shape errors we used the estimates of modulation loss through self-apodization and its uncertainties, as presented by Hase (2003).

Although a spectral shift correction is carried out in a step preceding the combined temperature and pointing retrieval (see, Section3.2), a residual frequency calibration error is considered. It is estimated as the root mean squares difference between the obtained frequency corrections from the shift retrievals and the linear regression line of these spectral shifts over wavenumber. The resulting uncertainty is 0.00029 cm$^{-1}$.

Uncertainties in pointing and radiance offset (zero calibration) were not explicitly considered in the error estimation, because these quantities were simultaneously retrieved with temperature.

### 4.1.2 Parameter errors

During the temperature retrieval, the concentrations of all interfering gases except O$_3$ and H$_2$O are assumed to be known and treated as parameters. In preceding MIPAS retrievals, climatological distributions of these interfering gases were used for this purpose. Accordingly, the climatological variability determined the uncertainty. For MIPAS version 8 retrievals, results from preceding MIPAS data processing were already available and could be used as estimates of the actual concentrations. The respective uncertainties reduce to the uncertainties of the preceding retrieval. Resulting temperature uncertainties are below 0.1 K for all interfering species that were not jointly fitted.

Since CO$_2$ lines are used for the temperature retrieval, results are deemed particularly sensitive to the assumed CO$_2$ mixing ratios. These were taken from the WACCM4 runs described in Section 3.4. Respective estimated 1-$\sigma$ uncertainties are reported in Table 3. In the troposphere and stratosphere, these are based on considerations of CO$_2$ uncertainties according to the IPCC Fifth assessment report and uncertainties due the seasonal variability of CO$_2$. Above the middle mesosphere, they were estimated after comparisons between WACCM CO$_2$ and measurements from space, mainly SABER and ACE, as shown in López-Puertas et al. (2017).





**Table 3.** CO$_2$ mixing ratio uncertainties

| Altitude | Uncertainty |
| --- | --- |
| below 30 km | 0.2% |
| 40 km | 0.5% |
| 60 km | 1.0% |
| 64-80 km | 2.0% |
| 90 km | 10% |
| 100 km | 10% |
| 120 km | 30% |

### 4.1.3 Model errors

Since the true atmospheric radiative transfer is not known, genuine model deficiencies could not be quantified. However, past model intercomparisons (von Clarmann et al., 2001, 2003b; Tjemkes et al., 2003; Schreier et al., 2018) do not hint at any obvious malfunction. Numerical accuracy has been tuned to a degree that corresponding temperature errors are insignificant compared to the leading error sources. The most problematic error source in the category of model errors is the uncertainty of spectroscopic data.

The uncertainty estimates can vary considerably, and it is not always clear what they represent. Additionally there is little information available, whether the uncertainties of different spectral lines and/or bands are correlated or not. After consultation by a laboratory spectroscopist (Manfred Birk, personal communication, February 2020) we use the following 1-$\sigma$ uncertainty estimates: CO$_2$ line intensities: 1%; pressure broadening coefficients: 2%; exponent for temperature dependence: 0.2 (absolute).

We further assume that these uncertainties are fully correlated between different lines. We concede that these assumptions 400 can be challenged. However, since we report the temperature uncertainties caused by spectroscopic uncertainties separately, data users endowed with a different degree of optimism can easily rescale the resulting error estimates. Uncertainty estimates provided along with the spectroscopic data compilation by Flaud et al. (2003) appear to be less optimistic than ours. However, preliminary validation do not support the hence resulting larger temperature bias.

For former MIPAS temperature data, uncertainties due to the neglect of non-linear thermodynamic equilibrium and unac-405 counted horizontal variability of the atmospheric state were reported. These error sources are not considered here, because non-linear thermodynamic equilibrium effects and the horizontally varying atmosphere are explicitly modeled (see Sections 3.11 and 3.5).

### 4.2 Random errors

Random errors are errors which explain the standard deviation of the differences between measurements of the same state 410 variable by two different instruments. The main sources of random errors of MIPAS temperature are measurement noise, spectral shift, gain calibration uncertainties, and the uncertainties of CO$_2$ mixing ratios. Measurement noise is random by





its nature. Spectral shift has originally a more systematic characteristic, but the residual frequency calibration error after correction is random. According to our definition, also gain calibration uncertainties are random. While they are obviously systematic within one gain calibration period, they contribute in the long run rather to the standard deviation of differences

between measurement systems than to the bias. Similar considerations apply to the uncertainties in $CO_2$ mixing ratios, which we consider as random, although they are presumably positively correlated among subsequent measurements. The adequacy of this classification of uncertainties in random and systematic components will be critically tested in a dedicated validation study. None of the other random error components, e.g., mixing ratios of interfering species, makes a sizeable contribution to the error budget.

For most atmospheric conditions and altitudes, the random temperature uncertainty varies between 0.4 and 0.8 K. Occasional excursions up to 1.3 K are encountered above 60 km altitude (Tables A1–A9 and Figures 1–2).

As a rule of thumb, measurement noise is – everything else unchanged – larger for colder and smaller for warmer atmospheres. For the other random error components, no such simple dependence of the error on the atmospheric state can be provided.

For some applications the error covariances are relevant. These depend both on the structure of the Jacobian of the inverse problem and on the covariances of the ingoing uncertainties. While it is hard to fully quantify the latter, we present a sample error correlation matrix which characterises the former in the Appendix (Table B1). The correlation matrix allows the construction of an approximate covariance matrix for any given retrieval noise.

## 4.3 Systematic errors

Systematic errors are, regardless of their origin, errors which explain the bias between measurements of the same state variable by different instruments observing the same part of the atmosphere. The main sources of systematic error in MIPAS temperatures below the mid-mesosphere are uncertainties in spectroscopic data and instrument line shape uncertainties. To classify these as systematic is admittedly an idealization, because the actual conditions will somehow modulate the actual resulting errors; e.g., the impact of the uncertainty of the line intensity of an interfering gas depends on the abundance of the interfering

species, which may vary randomly. Since, however, the $CO_2$ lines chosen for the temperature retrieval are strong lines and only weakly interfered by transitions of other species, this random modulation of systematic errors is deemed negligible and the classification of related temperature uncertainties as chiefly systematic seems justified.

The other source of systematic error in MIPAS temperatures is uncertainties in the instrument line shape. Since the same set of coefficients is used for all measurements, this error is of clearly systematic nature. However, it must be kept in mind that

modulations of the related initially systematic error by the variable sensitivity of the retrieval that depends on the actual state of the atmosphere will generate a certain random component.

In all altitudes except the uppermost ones, the error budget is dominated by these systematic errors. With this in mind, it can be considered as a particularly grave deficit that uncertainties in spectroscopic data are so vaguely characterized with respect to their confidence limits and correlation characteristics.



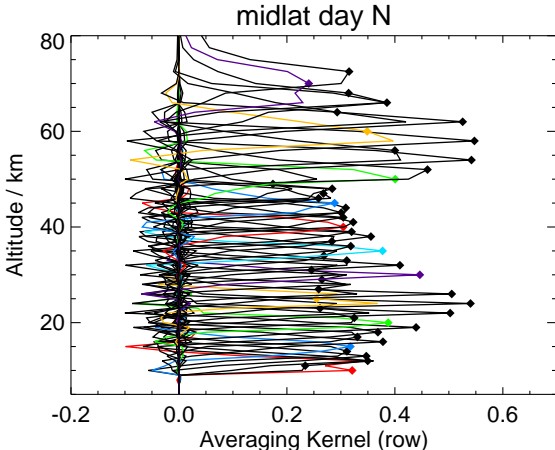

**Figure 4.** MIPAS temperature averaging kernels for a northern midlatitude daytime observation. For clarity the kernels belonging to retrieval altitudes 5, 10, 15, . . . km are color-coded. Diamonds represent the nominal altitudes.

### 4.4 Error correlations in various domains

Since our retrieval decomposes the inverse problem profile by profile, vertical correlations of measurement noise are represented by the respective covariance matrix. Related correlation coefficients are represented in Table B1. Errors due to spectral shift are expected to be almost fully correlated in the altitude domain because the frequency calibration correction is performed individually for entire limb scans. Since frequency calibration corrections are constrained towards the long-term mean, also a positive error correlation in the time domain has to be expected.

As stated above, positive correlations are expected for gain calibration errors of measurements recorded within one gain calibration period. This leads to positive error correlations in altitude and between subsequent limb scans. The typical length of a gain calibration period is one day, occasionally two days. Also errors due to uncertain mixing ratios of $CO_2$ are expected to be correlated in altitude and between subsequent limb scans. Correlation lengths depend on the actual spatial and temporal extension of the $CO_2$ anomalies.

## 5 Results

### 5.1 Averaging kernels and vertical resolution

The vertical resolution of the temperature profiles, estimated as the full width at half maximum of the respective row of the averaging kernel matrix, varies around 3 km in the altitude range up to 40 km (Fig. 4). Above, it gradually deteriorates towards 7 km at 70 km. A local maximum of vertical resolution values of approx. 3.3 km is typically found at the tropical tropopause layer (around 15 km altitude) and is attributed to particularly cold temperatures. The actual values of the vertical resolution are provided for each limb scan along with the data on the MIPAS data server (http://www.imk-asf.kit.edu/english/308.php).





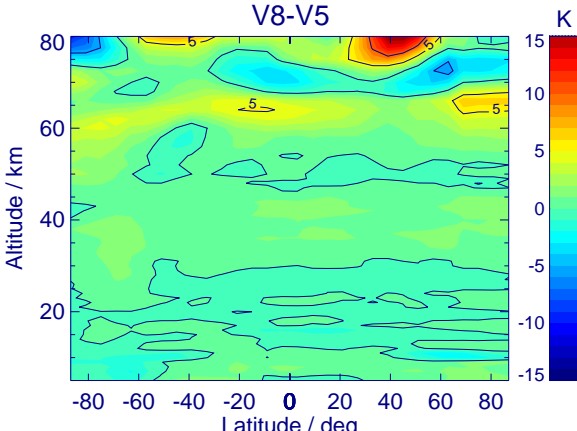

**Figure 5.** Mean monthly MIPAS temperature differences between version V8R_T_261 - V5R_T_220 for August 2010.

The averaging kernels are generally well-behaved in the sense that they peak at their nominal altitude. That is to say, the temperature retrieval at altitude $z$ is most sensitive to the true temperature at altitude $z$. Further, the kernels are fairly symmetric.

This rules out major information displacement by the retrieval. The pronounced side-wiggles are a typical feature of a retrieval on a grid that is much finer than the measurement grid. This does not point at a weakness of the retrieval set-up. Instead, the often smoother averaging kernels of retrievals on coarser retrieval grids just do not represent these features because the Jacobians do not resolve them. Understanding the column of an averaging kernel matrix as the response of the retrieval to a delta perturbation of the true profile, the so-called delta perturbation on a coarse grid perturbs a much wider part of the

atmosphere and thus is not comparable to our fine-grid averaging kernels.

### 5.2 Temperature differences with respect to previous data versions

Preceding versions of MIPAS temperature data were already quite a mature and well validated data product (e.g. Wang et al., 2004, 2005). It has already been shown that MIPAS sees the expected temperature features in the middle atmosphere (e.g., von Clarmann et al., 2009). Thus, it does not come as a surprise that for most parts of the atmosphere, the differences between

the new improved temperature data and the previous ones are small. (Fig. 5). Only near the stratopause and above major differences are observed. These are attributed to the use of the extended set of microwindows (see Section 3.6) and to the new WACCM-based prior information (see Section 3.4), which is expected to represent the actual conditions much better than the MSIS-based climatology used before.

In this Section we concentrate on improvements with respect to the previous data version for cases where problems with

the older data had already been identified, and we investigate, to which degree MIPAS provides additional information with respect to pre-existing knowledge on temperature and line-of-sight pointing.





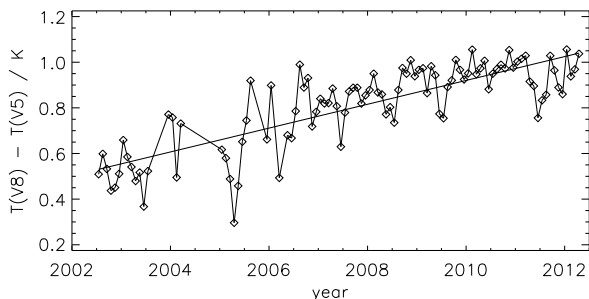

**Figure 6.** Time series of MIPAS temperature differences between version 8 and version 5 data, for the altitude range 35 to 40 km, averaged between 90°N and 90°S.

### 5.2.1 Drifts

The technical aspects of the drifts in MIPAS data due to detector aging have already been discussed in Section 2. Here we assess to which degree the revised non-linearity correction in the level-1b processing was successful to reduce related drifts in temperature. Figure 6 shows a time series of temperature differences between MIPAS version 8 and version 5 data. The altitude range of this example is 35 to 40 km and the latitudinal coverage is global. Results for other altitudes are similar.

A relative drift between the data sets is obvious, and comparison with the data by McLandress et al. (2015) clearly suggests that the temporal development of the MIPAS V8 data is more realistic than that of the V5 data. This means that the new MIPAS nonlinearity correction successfully reduces the negative temperature drift.

### 5.2.2 Consistency between high resolution and reduced resolution results

In time series of MIPAS V5 data products jumps in atmospheric state variables can be often seen between the full spectral resolution period (2002–2004) and the reduced spectral resolution period (2005-2012). Although methodical development work was never targeted at removing these jumps, as a side effect of other retrieval optimization work, the full resolution and the reduced resolution datasets have become much better interconsistent in the sense that these jumps are now largely reduced.

An illustration of this inconsistency problem is given in Fig. 7. The top row shows monthly temperature means of V8 data in 10° bins for FR (July 2003) and RR (July 2009) data. There is no obvious inconsistency. However, the lower row of Fig. 7 shows, that the differences between V8 and V5 monthly mean data clearly differ for the full resolution data (i.e. V8 minus V5 for FR measurement period, lower left) and the reduced resolution data (V8 minus V5 for RR, lower right).

To further clarify this inconsistency, the difference between reduced resolution and full resolution monthly mean data was calculated separately for data versions V5 and V8. From Figure 8 it is obvious, that the differences in V5 (left panel) are much more pronounced compared to those in V8. The structure of remaining differences in V8 can also be seen in the V5 differences, suggesting that this is a real atmospheric feature, since mean temperatures of July 2009 and 2003 can be expected to differ somewhat. The result of this analysis is that our V8 data is much more consistent between the MIPAS FR and RR measurement periods than preceding data versions.



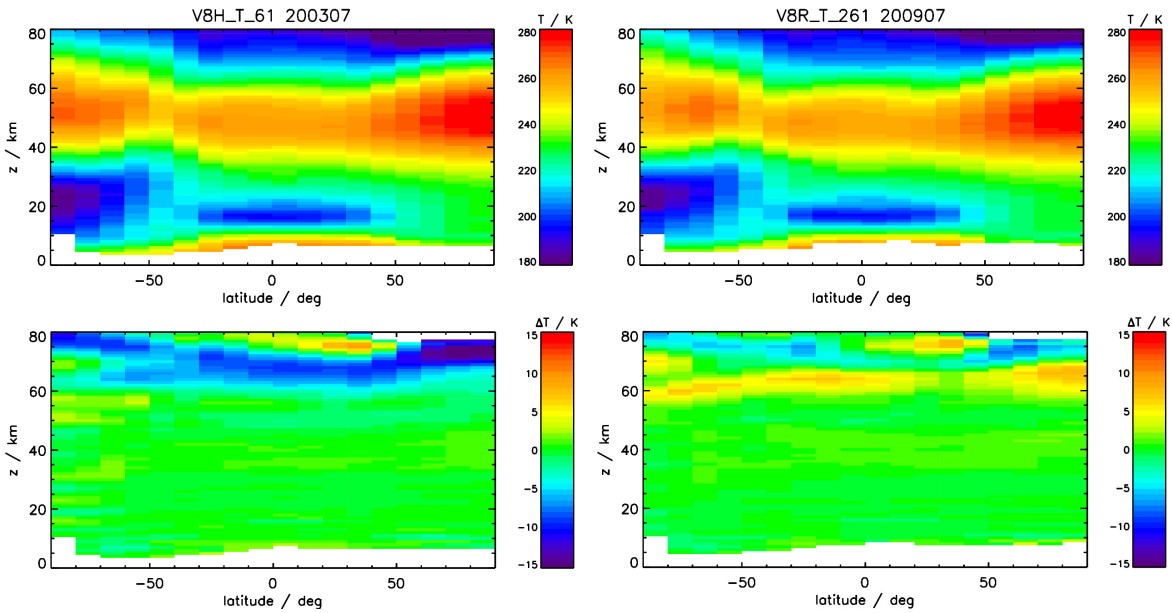

**Figure 7.** Upper row: Monthly mean V8 temperature in 10° latitude bins for July 2003 (FR period, left) and July 2009 (RR, right). Lower left: differences between V8 and V5 monthly mean temperatures for FR data (July 2003). Lower right: differences between V8 and V5 monthly mean temperatures for RR data (July 2009).

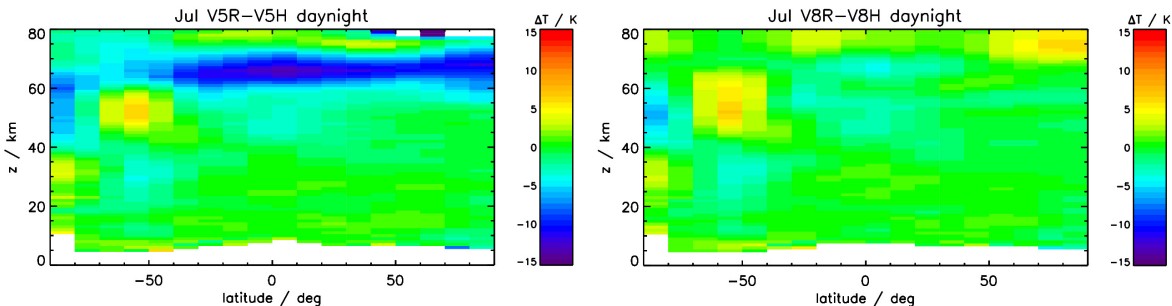

**Figure 8.** Difference between reduced resolution and full resolution July mean data for V5 (left) and V8 (right).





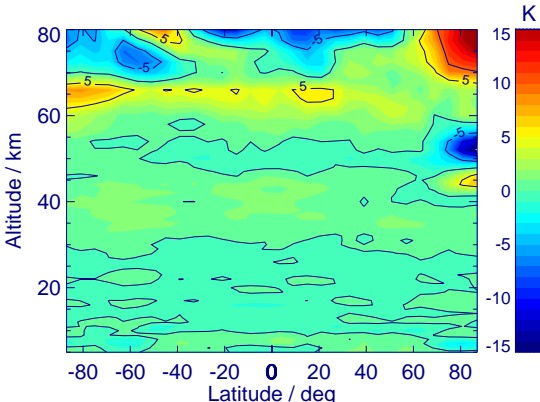

**Figure 9.** Difference between V8R_T_261 and V5R_T_220 temperature for 20 February 2009.

### 5.2.3 Case study: Elevated stratopause situations

The dependence of retrieved temperatures above about 60 km on the prior information is caused by the fact that MIPAS cannot resolve the shape of the temperature profile above the highest tangent altitude. This problem has motivated us to replace at these altitudes the climatological NRLMSISE-00-based prior information with prior information from a debiased specified dynamics WACCM run (see, Section 3.4). As a test case, an elevated stratopause event in February 2009 was chosen. A discussion of this episode and independent evidence of this event are reported, e.g., in Funke et al. (2017). The onset of this event was in the beginning of February, and in the second half of February the temperature anomaly reached altitudes relevant to MIPAS retrievals.

Figure 9 shows the difference between V8 and V5 temperature for February 20, 2009. The different behaviour of the retrievals is evident. Globally, differences in the data versions are confined to altitudes above 60 km and occasionally exceed 5 K. Here the positive temperature differences hint at too low temperatures in version 5 even at altitudes where MIPAS has measurement information. This is a result of error correlations with altitudes above about 68 km where the retrieval has to rely on the shape of the a priori profile. The too cold temperatures in V5 (showing up as positive differences V8-V5) compensate the too warm a priori-driven temperatures above 70 km to best fit the measured radiance signal.

At northern polar latitudes the inclusion of the new a priori information, which better reflects the actual conditions, is more drastic. The warm region above 70 km is not represented by the V5 NRLMSISE-00-based a priori, and this error propagates downward to 40 km, showing up as temperature oscillations with too warm temperatures in V5 (negative V8-V5 difference) around 50 km and too cold temperatures (positive V8-V5 differences) around 42 km altitude. In summary, the new data version better represents this event not only at altitudes above the uppermost tangent altitude (around 68 km) but also below, because the inadequate temperature profile above the uppermost tangent altitude in V5_T_221 triggered, via error correlations, temperature errors also at altitudes where MIPAS is able to resolve the temperature profile.



## 5.3 Case study on differences with respect to ECMWF temperatures: Temperature waves

MIPAS is able to reveal structures in temperature profiles independently of the a priori information. We demonstrate this by two examples of features in temperature profiles, which might be attributed to gravity waves. In the left panels of Fig. 10 temperature profiles for MIPAS retrievals (black lines) and the corresponding ECMWF-based a priori (red lines) are shown,
while the right panels show the differences of temperature profile minus the respective vertically smoothed temperature profile for retrieval and a priori data. Smoothing is done with a boxcar of 10 km width.

We rule out that the retrieved wave structure is a numerical artefact of the retrieval caused by too weak regularization, because the MIPAS result agrees well with the ECMWF ERA-interim analysis, which shows very similar structure. The upper panels in Figure 10 show an example. The example shown in the lower panel demonstrates, that MIPAS is able to retrieve
such structures independently from the a priori information. There (as in many other cases) we find wave structures in both datasets, MIPAS and ECMWF analyses, with similar vertical wavelength but different phase. The retrieval scheme chosen does not employ any mechanism that would be able to map a vertically shifted structure in the prior information onto the result. Therefore these results prove that structures in vertical profiles, and in particular these wave structures, are independent MIPAS measurement information.

## 5.4 Pointing differences with respect to level-1b engineering information

Contrary to other MIPAS data processors (Dinelli et al. 2010, Raspollini et al. 2013 and (http://www.atm.ox.ac.uk/MORSE/), the IMK/IAA processor retrieves the pointing information in terms of tangent altitudes from the spectra, using the engineering information as a Bayesian constraint only, but not as a hard constraint (see Section 3.3). The comparison between the retrieved data and the level-1b engineering information has been used in the past to characterize the MIPAS pointing, and to improve
the algorithm involved in the calculation of the line of sight (Kiefer et al., 2007). Meanwhile several improvements of this algorithm have been implemented, and now the comparison reveals the following:

1. The engineering information on the tangent altitudes has changed in a noticeable manner between data versions V5 and V8. Mean changes between engineering tangent altitudes exceeded 600 m at most altitudes.

2. Mean differences in retrieved tangent altitudes (V8-V5) are smaller than about 100 m at altitudes below 40 km and
steadily increase above to values of 600 m at 60 km altitude.

3. Differences between engineering tangent altitudes and retrieved tangent altitudes are largest for data version V5. They vary around 400 m in large parts of the altitude range, and engineering tangent altitudes are larger than the retrieved ones. Smallest differences are found for data version V8. V8 engineering tangent altitudes are, on average, lower than the retrieved ones by about 200 m below 40 km and by about 50 m above. This suggests that the engineering tangent
altitudes have improved considerably over the versions.





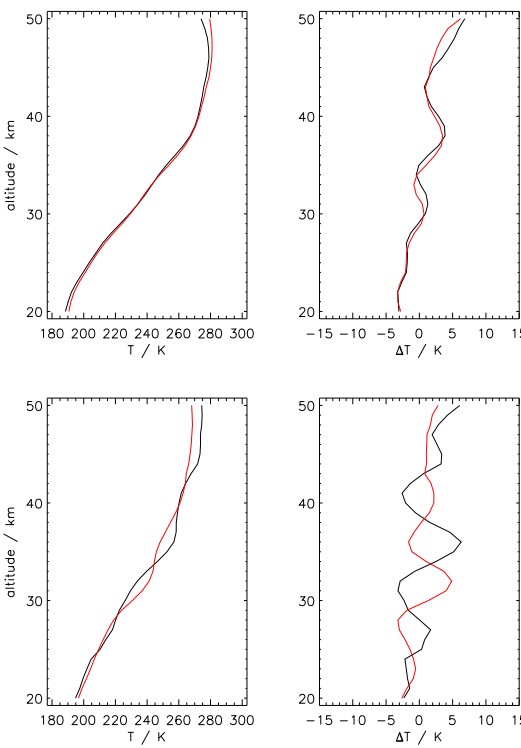

**Figure 10.** Temperature profiles (left column) and temperature profile minus altitude-smoothed temperature profile (right) at 67.3°S, 32.5°W on September 24, 2009 (upper panels) and at 62.0°S 5.6°W on September 5, 2009 (lower panels) for MIPAS (black curves) and ECMWF ERA-Interim (red). MIPAS and ECMWF agree well with respect to the wavelength of the temperature waves, but not always with respect to the phase.

4. The standard deviations of the differences beween engineering tangent altitudes and retrieved tangent altitudes was reduced from about 600 m in version 5 to about 150 m in version 8. Also this indicates an improvement of the level-1b tangent altitudes.

5. No discernable latitude dependence was found in these differences.

6. These results confirm that indeed quite independent tangent altitude information is retrieved by the IMK/IAA MIPAS processor and that the retrieval is not over-constrained towards the engineering information.



## 6 Conclusions

In summary, the retrieval vector of the IMK/IAA temperature retrieval contains, besides the temperature profile, (a) linear horizontal gradients in latitude and longitude directions, (b) tangent altitudes of all spectra of the limb scan under analysis, (c–d) vertical profiles of $O_3$ and $H_2O$ mixing ratios, (e) a vertical profile of background continuum emission per microwindow, and (f) a radiance offset correction for each microwindow and each tangent altitude. Beyond new level-1b radiance spectra, improvements with respect to older data versions refer to the following upgrades of the retrieval scheme: The frequency calibration correction scheme is made more robust. Additional microwindows were included to obtain more information from high altitudes. A non-LTE parameterization that accounts for the temperature dependence of vibrational non-LTE populations in an approximate manner has been adopted. Better temperature a priori information is used for higher altitudes, taking the actual conditions better into account. Trace gas mixing ratios from previous MIPAS data versions are used to model the contributions of interfering species. An empirical background continuum is retrieved to altitudes up to 58 km instead of 32 km only. An improved offset calibration correction has been used. Due to their significant contribution to the signal in $CO_2$ microwindows, mixing ratios of $O_3$ and $H_2O$ were jointly fitted. Forward calculations were based on updated spectroscopic data. A TUNER-compliant error budget is provided.

The developments described above led to the following improvements in the MIPAS temperature data: The drift caused by the non-linearity correction applied in the course of the radiance calibration has been reduced. Results from the high spectral resolution period (2002–2004) and the reduced spectral resolution period (2005-2012) are now more consistent. Temperature profiles for situations where the temperature profile above the altitude range covered by MIPAS tangent altitudes deviates strongly from the climatological mean, e.g., elevated stratopause situations, are now much more realistic. Compared to previous data versions, a larger fraction of the retrievals converged. We have shown that, although ECMWF ERA-Interim temperature fields are used to constrain the temperature retrievals, vertical temperature wave information can be retrieved which is independent of the prior information used.

The further evaluation of MIPAS version 8 temperatures is deferred to a dedicated validation study. This work is confined to measurements recorded in nominal and UTLS measurement modes. The temperature retrieval from spectra recorded in the middle and upper atmospheric measurement modes are reported in a companion paper by García-Comas et al. (2020).

The MIPAS data can be obtained from the IMK/IAA MIPAS data server under https://www.imk-asf.kit.edu/english/308.php.

*Acknowledgements.* Spectra used for this work were provided by the European Space Agency. We would like to thank the MIPAS Quality Working Group for enlightening discussions, Claus Zehner for particularly helpful support. This study was partly funded by DLR under contract number 50EE1547 (SEREMISA). The IAA team was supported by MCIU under projects ESP2017-87143-R and PID2019-110689RB-I00/AEI/10.13039/501100011033. The computations were done in the frame of a Bundesprojekt (grant MIPAS_V7) on the Cray XC40 "Hazel Hen" of the High-Performance Computing Center Stuttgart (HLRS) of the University of Stuttgart. WACCM simulations are based upon work supported by the National Center for Atmospheric Research (NCAR), which is a major facility sponsored by the National Science





**Table A1.** Temperature error budget for northern polar winter. All uncertainties are $1\sigma$.

| Altitude | Temp. | Total Error | Random Error | Syst. Error | Meas. Noise | Gain Calibr. | Spectral Shift | $CO_2$-VMR | Spectrosc. Data | Instrument Line Shape |
|---|---|---|---|---|---|---|---|---|---|---|
| (km) | (K) | (K) | (K) | (K) | (K) | (K) | (K) | (K) | (K) | (K) |
| 9 | 216.1 | 1.1 | 0.6 | 0.9 | 0.5 | 0.3 | <0.1 | <0.1 | 0.8 | 0.5 |
| 12 | 210.6 | 1.1 | 0.5 | 0.9 | 0.4 | 0.3 | <0.1 | <0.1 | 0.8 | 0.5 |
| 15 | 205.6 | 1.5 | 0.5 | 1.4 | 0.4 | 0.3 | <0.1 | <0.1 | 1.2 | 0.7 |
| 18 | 199.6 | 1.1 | 0.5 | 1.0 | 0.4 | 0.3 | <0.1 | <0.1 | 0.9 | 0.4 |
| 21 | 194.2 | 0.8 | 0.5 | 0.6 | 0.3 | 0.3 | <0.1 | <0.1 | 0.5 | 0.3 |
| 24 | 192.6 | 0.5 | 0.4 | 0.2 | 0.3 | 0.3 | <0.1 | <0.1 | 0.2 | 0.1 |
| 27 | 195.6 | 0.8 | 0.5 | 0.6 | 0.4 | 0.3 | <0.1 | <0.1 | 0.6 | 0.1 |
| 30 | 203.7 | 1.0 | 0.4 | 0.9 | 0.3 | 0.3 | <0.1 | <0.1 | 0.8 | 0.3 |
| 33 | 213.2 | 1.1 | 0.5 | 1.0 | 0.3 | 0.3 | 0.1 | <0.1 | 0.9 | 0.4 |
| 36 | 223.0 | 0.9 | 0.5 | 0.7 | 0.3 | 0.3 | <0.1 | <0.1 | 0.7 | 0.2 |
| 39 | 230.5 | 0.8 | 0.5 | 0.6 | 0.3 | 0.4 | <0.1 | <0.1 | 0.6 | 0.2 |
| 42 | 237.8 | 0.6 | 0.5 | 0.4 | 0.3 | 0.4 | <0.1 | <0.1 | 0.3 | 0.2 |
| 45 | 241.2 | 0.7 | 0.5 | 0.4 | 0.4 | 0.4 | 0.1 | <0.1 | 0.2 | 0.3 |
| 48 | 245.3 | 0.8 | 0.6 | 0.5 | 0.4 | 0.4 | <0.1 | <0.1 | 0.2 | 0.5 |
| 52 | 251.7 | 0.9 | 0.7 | 0.6 | 0.6 | 0.4 | <0.1 | <0.1 | 0.3 | 0.6 |
| 56 | 255.2 | 1.0 | 0.7 | 0.7 | 0.6 | 0.4 | <0.1 | <0.1 | 0.4 | 0.6 |
| 60 | 252.9 | 1.2 | 0.8 | 0.8 | 0.7 | 0.4 | <0.1 | <0.1 | 0.5 | 0.6 |
| 64 | 248.5 | 1.3 | 0.8 | 1.0 | 0.7 | 0.4 | <0.1 | 0.1 | 0.7 | 0.6 |
| 68 | 237.9 | 1.6 | 1.1 | 1.1 | 1.0 | 0.4 | 0.1 | 0.1 | 0.9 | 0.6 |

Foundation under Cooperative Agreement No. 1852977. WACCM computing resources were provided by the Climate Simulation Laboratory
at NCAR's Computational and Information Systems Laboratory.

**Appendix A:  Representative errors**

The error budget of MIPAS temperatures for nine representative atmospheric conditions is reported in Tables A1–A9. The atmospheric conditions under consideration are northern and southern polar winter, polar summer, northern and southern midlatitudes for day and night, and tropical day and night.





**Table A2.** Temperature error budget for southern polar winter. All uncertainties are $1\sigma$.

| Altitude | Temp. | Total Error | Random Error | Syst. Error | Meas. Noise | Gain Calibr. | Spectral Shift | $CO_2$-VMR | Spectrosc. Data | Instrument Line Shape |
|---|---|---|---|---|---|---|---|---|---|---|
| (km) | (K) | (K) | (K) | (K) | (K) | (K) | (K) | (K) | (K) | (K) |
| 24 | 183.1 | 1.0 | 0.7 | 0.6 | 0.7 | 0.2 | <0.1 | <0.1 | 0.6 | 0.1 |
| 27 | 187.4 | 1.1 | 0.6 | 0.9 | 0.6 | 0.2 | <0.1 | <0.1 | 0.9 | 0.1 |
| 30 | 197.8 | 1.3 | 0.5 | 1.1 | 0.5 | 0.2 | <0.1 | <0.1 | 1.1 | 0.3 |
| 33 | 208.9 | 1.6 | 0.5 | 1.5 | 0.5 | 0.3 | 0.1 | <0.1 | 1.4 | 0.4 |
| 36 | 226.4 | 1.6 | 0.6 | 1.5 | 0.4 | 0.4 | <0.1 | <0.1 | 1.5 | 0.1 |
| 39 | 241.3 | 1.2 | 0.6 | 1.0 | 0.4 | 0.4 | <0.1 | <0.1 | 1.0 | 0.2 |
| 42 | 252.7 | 0.9 | 0.6 | 0.6 | 0.4 | 0.4 | 0.1 | <0.1 | 0.6 | 0.2 |
| 45 | 260.3 | 0.7 | 0.6 | 0.4 | 0.4 | 0.4 | 0.1 | <0.1 | 0.3 | 0.3 |
| 48 | 263.8 | 0.9 | 0.6 | 0.6 | 0.4 | 0.4 | <0.1 | <0.1 | 0.2 | 0.5 |
| 52 | 265.9 | 1.1 | 0.7 | 0.8 | 0.6 | 0.4 | <0.1 | <0.1 | 0.5 | 0.6 |
| 56 | 258.7 | 1.3 | 0.8 | 0.9 | 0.7 | 0.4 | <0.1 | <0.1 | 0.6 | 0.7 |
| 60 | 253.4 | 1.4 | 0.9 | 1.0 | 0.9 | 0.4 | <0.1 | <0.1 | 0.7 | 0.8 |
| 64 | 244.9 | 1.4 | 1.0 | 1.0 | 0.9 | 0.3 | 0.1 | 0.1 | 0.7 | 0.7 |
| 68 | 239.9 | 1.4 | 1.2 | 0.8 | 1.1 | 0.3 | 0.1 | 0.2 | 0.6 | 0.6 |

**Appendix B: Noise error correlations**

For some applications the error covariances are relevant. In Table B1 we present a sample error correlation matrix. The error correlation matrix is a covariance matrix of retrieval noise component-wise divided by the standard deviations. The result is a matrix of correlation components that can be used to construct an approximate covariance matrix for any given retrieval noise.



**Table A3.** Temperature error budget for polar summer. All uncertainties are $1\sigma$.

| Altitude | Temp. | Total Error | Random Error | Syst. Error | Meas. Noise | Gain Calibr. | Spectral Shift | $CO_2$- VMR | Spectrosc. Data | Instrument Line Shape |
|---|---|---|---|---|---|---|---|---|---|---|
| (km) | (K) | (K) | (K) | (K) | (K) | (K) | (K) | (K) | (K) | (K) |
| 9 | 216.8 | 1.7 | 0.6 | 1.6 | 0.5 | 0.4 | <0.1 | <0.1 | 1.2 | 1.0 |
| 12 | 218.0 | 0.6 | 0.4 | 0.4 | 0.3 | 0.2 | <0.1 | <0.1 | 0.3 | 0.3 |
| 15 | 225.3 | 0.9 | 0.4 | 0.8 | 0.3 | 0.2 | <0.1 | <0.1 | 0.6 | 0.6 |
| 18 | 234.0 | 0.7 | 0.4 | 0.6 | 0.2 | 0.3 | <0.1 | <0.1 | 0.3 | 0.5 |
| 21 | 236.9 | 0.5 | 0.5 | 0.3 | 0.2 | 0.4 | <0.1 | <0.1 | 0.1 | 0.2 |
| 24 | 239.7 | 0.5 | 0.5 | 0.3 | 0.3 | 0.4 | <0.1 | <0.1 | 0.2 | 0.2 |
| 27 | 241.6 | 0.5 | 0.4 | 0.3 | 0.2 | 0.4 | 0.1 | <0.1 | 0.2 | 0.2 |
| 30 | 244.9 | 0.7 | 0.4 | 0.6 | 0.3 | 0.3 | <0.1 | <0.1 | 0.3 | 0.5 |
| 33 | 250.2 | 1.1 | 0.4 | 1.0 | 0.3 | 0.3 | 0.1 | <0.1 | 0.5 | 0.8 |
| 36 | 256.7 | 0.8 | 0.5 | 0.6 | 0.3 | 0.4 | 0.2 | <0.1 | 0.5 | 0.4 |
| 39 | 265.3 | 0.8 | 0.6 | 0.6 | 0.3 | 0.5 | <0.1 | <0.1 | 0.6 | 0.2 |
| 42 | 274.0 | 0.8 | 0.6 | 0.5 | 0.3 | 0.6 | 0.1 | <0.1 | 0.5 | 0.1 |
| 45 | 282.1 | 0.7 | 0.6 | 0.3 | 0.3 | 0.5 | 0.1 | <0.1 | 0.3 | 0.1 |
| 48 | 284.4 | 0.6 | 0.6 | 0.3 | 0.3 | 0.5 | <0.1 | <0.1 | 0.1 | 0.3 |
| 52 | 284.9 | 0.9 | 0.6 | 0.7 | 0.4 | 0.4 | 0.1 | <0.1 | 0.3 | 0.6 |
| 56 | 278.6 | 1.1 | 0.6 | 0.9 | 0.5 | 0.4 | 0.1 | <0.1 | 0.6 | 0.7 |
| 60 | 268.8 | 1.3 | 0.7 | 1.1 | 0.6 | 0.4 | <0.1 | <0.1 | 0.8 | 0.7 |
| 64 | 253.1 | 1.6 | 1.0 | 1.2 | 0.6 | 0.8 | <0.1 | <0.1 | 1.1 | 0.5 |
| 68 | 235.3 | 1.8 | 1.1 | 1.4 | 0.8 | 0.7 | <0.1 | 0.1 | 1.2 | 0.8 |



**Table A4.** Temperature error budget for northern midlatitudes, day. All uncertainties are $1\sigma$.

| Altitude | Temp. | Total Error | Random Error | Syst. Error | Meas. Noise | Gain Calibr. | Spectral Shift | $CO_2$-VMR | Spectrosc. Data | Instrument Line Shape |
|---|---|---|---|---|---|---|---|---|---|---|
| (km) | (K) | (K) | (K) | (K) | (K) | (K) | (K) | (K) | (K) | (K) |
| 12 | 224.4 | 2.3 | 0.8 | 2.1 | 0.6 | 0.6 | <0.1 | <0.1 | 1.8 | 1.1 |
| 15 | 213.8 | 1.3 | 0.5 | 1.1 | 0.4 | 0.3 | <0.1 | <0.1 | 0.8 | 0.9 |
| 18 | 216.5 | 0.6 | 0.4 | 0.3 | 0.3 | 0.3 | <0.1 | <0.1 | 0.3 | 0.2 |
| 21 | 219.2 | 0.6 | 0.4 | 0.4 | 0.2 | 0.3 | <0.1 | <0.1 | 0.3 | 0.2 |
| 24 | 224.9 | 0.6 | 0.4 | 0.4 | 0.3 | 0.3 | <0.1 | <0.1 | 0.3 | 0.1 |
| 27 | 229.5 | 0.7 | 0.4 | 0.5 | 0.3 | 0.3 | <0.1 | <0.1 | 0.4 | 0.2 |
| 30 | 234.9 | 0.8 | 0.4 | 0.6 | 0.3 | 0.3 | <0.1 | <0.1 | 0.4 | 0.4 |
| 33 | 239.5 | 0.9 | 0.5 | 0.8 | 0.3 | 0.3 | 0.1 | <0.1 | 0.5 | 0.6 |
| 36 | 245.9 | 0.8 | 0.5 | 0.7 | 0.3 | 0.3 | 0.1 | <0.1 | 0.5 | 0.4 |
| 39 | 253.2 | 0.9 | 0.5 | 0.7 | 0.3 | 0.5 | <0.1 | <0.1 | 0.7 | 0.2 |
| 42 | 263.1 | 0.9 | 0.6 | 0.6 | 0.3 | 0.5 | 0.1 | <0.1 | 0.6 | 0.2 |
| 45 | 269.4 | 0.7 | 0.6 | 0.3 | 0.3 | 0.5 | <0.1 | <0.1 | 0.2 | 0.3 |
| 48 | 270.2 | 0.7 | 0.6 | 0.4 | 0.4 | 0.4 | <0.1 | <0.1 | 0.3 | 0.3 |
| 52 | 265.2 | 1.0 | 0.6 | 0.8 | 0.5 | 0.4 | 0.1 | <0.1 | 0.5 | 0.5 |
| 56 | 261.1 | 1.2 | 0.6 | 1.0 | 0.5 | 0.4 | <0.1 | <0.1 | 0.6 | 0.8 |
| 60 | 247.7 | 1.6 | 0.8 | 1.4 | 0.6 | 0.6 | <0.1 | <0.1 | 1.1 | 0.8 |
| 64 | 230.1 | 1.7 | 0.9 | 1.4 | 0.7 | 0.6 | <0.1 | 0.1 | 1.2 | 0.7 |
| 68 | 213.6 | 1.6 | 1.1 | 1.1 | 1.0 | 0.5 | <0.1 | 0.1 | 0.9 | 0.6 |



**Table A5.** Temperature error budget for southern midlatitudes, day. All uncertainties are $1\sigma$.

| Altitude | Temp. | Total Error | Random Error | Syst. Error | Meas. Noise | Gain Calibr. | Spectral Shift | $CO_2$-VMR | Spectrosc. Data | Instrument Line Shape |
|---|---|---|---|---|---|---|---|---|---|---|
| (km) | (K) | (K) | (K) | (K) | (K) | (K) | (K) | (K) | (K) | (K) |
| 9 | 215.8 | 2.0 | 0.8 | 1.9 | 0.5 | 0.6 | <0.1 | <0.1 | 1.7 | 0.7 |
| 12 | 212.6 | 0.7 | 0.5 | 0.4 | 0.4 | 0.3 | <0.1 | <0.1 | 0.3 | 0.3 |
| 15 | 211.7 | 1.1 | 0.5 | 1.0 | 0.4 | 0.3 | <0.1 | <0.1 | 0.7 | 0.7 |
| 18 | 207.3 | 0.8 | 0.5 | 0.7 | 0.3 | 0.4 | <0.1 | <0.1 | 0.6 | 0.3 |
| 21 | 203.8 | 0.6 | 0.5 | 0.4 | 0.3 | 0.4 | <0.1 | <0.1 | 0.3 | 0.2 |
| 24 | 201.1 | 0.6 | 0.5 | 0.3 | 0.3 | 0.4 | <0.1 | <0.1 | 0.2 | 0.2 |
| 27 | 199.2 | 0.6 | 0.5 | 0.3 | 0.3 | 0.3 | <0.1 | <0.1 | 0.2 | 0.2 |
| 30 | 203.1 | 0.9 | 0.5 | 0.7 | 0.4 | 0.3 | <0.1 | <0.1 | 0.6 | 0.4 |
| 33 | 208.6 | 0.9 | 0.5 | 0.7 | 0.4 | 0.3 | 0.1 | <0.1 | 0.6 | 0.3 |
| 36 | 215.9 | 1.1 | 0.5 | 1.0 | 0.3 | 0.4 | <0.1 | <0.1 | 1.0 | 0.1 |
| 39 | 227.4 | 1.0 | 0.5 | 0.9 | 0.3 | 0.4 | 0.1 | <0.1 | 0.9 | 0.1 |
| 42 | 235.4 | 1.0 | 0.6 | 0.8 | 0.4 | 0.4 | <0.1 | <0.1 | 0.8 | 0.1 |
| 45 | 248.5 | 1.0 | 0.6 | 0.8 | 0.4 | 0.5 | <0.1 | <0.1 | 0.8 | 0.2 |
| 48 | 257.0 | 0.8 | 0.6 | 0.5 | 0.4 | 0.4 | <0.1 | <0.1 | 0.3 | 0.4 |
| 52 | 260.1 | 1.2 | 0.7 | 0.8 | 0.6 | 0.4 | <0.1 | <0.1 | 0.6 | 0.6 |
| 56 | 253.3 | 1.3 | 0.8 | 1.0 | 0.7 | 0.4 | <0.1 | <0.1 | 0.7 | 0.6 |
| 60 | 248.0 | 1.3 | 0.9 | 0.9 | 0.8 | 0.4 | <0.1 | <0.1 | 0.5 | 0.7 |
| 64 | 239.9 | 1.5 | 1.0 | 1.1 | 0.9 | 0.4 | <0.1 | 0.1 | 0.8 | 0.7 |
| 68 | 225.1 | 1.7 | 1.3 | 1.1 | 1.2 | 0.3 | <0.1 | 0.1 | 1.0 | 0.6 |



**Table A6.** Temperature error budget for northern midlatitudes, night. All uncertainties are $1\sigma$.

| Altitude | Temp. | Total Error | Random Error | Syst. Error | Meas. Noise | Gain Calibr. | Spectral Shift | $CO_2$- VMR | Spectrosc. Data | Instrument Line Shape |
|---|---|---|---|---|---|---|---|---|---|---|
| (km) | (K) | (K) | (K) | (K) | (K) | (K) | (K) | (K) | (K) | (K) |
| 9 | 242.6 | 2.3 | 0.8 | 2.1 | 0.6 | 0.6 | <0.1 | <0.1 | 1.8 | 1.1 |
| 12 | 222.7 | 1.8 | 0.7 | 1.6 | 0.5 | 0.5 | <0.1 | <0.1 | 1.3 | 0.9 |
| 15 | 217.8 | 1.2 | 0.5 | 1.0 | 0.4 | 0.3 | <0.1 | <0.1 | 0.6 | 0.8 |
| 18 | 219.2 | 0.6 | 0.4 | 0.4 | 0.3 | 0.3 | <0.1 | <0.1 | 0.2 | 0.4 |
| 21 | 221.2 | 0.5 | 0.4 | 0.3 | 0.3 | 0.3 | <0.1 | <0.1 | 0.1 | 0.3 |
| 24 | 223.7 | 0.5 | 0.5 | 0.2 | 0.3 | 0.4 | <0.1 | <0.1 | 0.2 | 0.1 |
| 27 | 227.9 | 0.6 | 0.4 | 0.4 | 0.3 | 0.3 | <0.1 | <0.1 | 0.3 | 0.2 |
| 30 | 231.2 | 0.7 | 0.4 | 0.6 | 0.3 | 0.3 | <0.1 | <0.1 | 0.3 | 0.5 |
| 33 | 237.5 | 1.1 | 0.4 | 0.9 | 0.3 | 0.3 | <0.1 | <0.1 | 0.8 | 0.5 |
| 36 | 248.4 | 0.9 | 0.4 | 0.8 | 0.3 | 0.3 | 0.1 | <0.1 | 0.7 | 0.3 |
| 39 | 256.6 | 0.8 | 0.5 | 0.6 | 0.3 | 0.5 | <0.1 | <0.1 | 0.5 | 0.2 |
| 42 | 263.8 | 0.8 | 0.6 | 0.5 | 0.3 | 0.5 | 0.1 | <0.1 | 0.5 | 0.1 |
| 45 | 271.0 | 0.7 | 0.6 | 0.3 | 0.3 | 0.5 | <0.1 | <0.1 | 0.2 | 0.2 |
| 48 | 273.0 | 0.7 | 0.5 | 0.4 | 0.3 | 0.4 | <0.1 | <0.1 | 0.2 | 0.3 |
| 52 | 268.6 | 1.0 | 0.6 | 0.8 | 0.5 | 0.4 | 0.1 | <0.1 | 0.6 | 0.6 |
| 56 | 260.9 | 1.2 | 0.6 | 1.0 | 0.5 | 0.4 | <0.1 | <0.1 | 0.7 | 0.7 |
| 60 | 250.2 | 1.4 | 0.8 | 1.1 | 0.6 | 0.5 | <0.1 | <0.1 | 0.9 | 0.7 |
| 64 | 235.9 | 1.6 | 0.9 | 1.3 | 0.7 | 0.6 | <0.1 | 0.1 | 1.1 | 0.7 |
| 68 | 218.4 | 1.8 | 1.2 | 1.3 | 1.0 | 0.6 | <0.1 | 0.1 | 1.1 | 0.7 |





**Table A7.** Temperature error budget for southern midlatitudes, night. All uncertainties are $1\sigma$.

| Altitude | Temp. | Total Error | Random Error | Syst. Error | Meas. Noise | Gain Calibr. | Spectral Shift | $CO_2$-VMR | Spectrosc. Data | Instrument Line Shape |
|---|---|---|---|---|---|---|---|---|---|---|
| (km) | (K) | (K) | (K) | (K) | (K) | (K) | (K) | (K) | (K) | (K) |
| 12 | 206.4 | 0.8 | 0.5 | 0.5 | 0.5 | 0.3 | <0.1 | <0.1 | 0.4 | 0.3 |
| 15 | 209.3 | 0.9 | 0.5 | 0.7 | 0.4 | 0.2 | <0.1 | <0.1 | 0.5 | 0.5 |
| 18 | 206.9 | 0.8 | 0.4 | 0.7 | 0.3 | 0.3 | <0.1 | <0.1 | 0.6 | 0.3 |
| 21 | 203.9 | 0.6 | 0.4 | 0.5 | 0.3 | 0.3 | <0.1 | <0.1 | 0.4 | 0.2 |
| 24 | 201.4 | 0.6 | 0.5 | 0.3 | 0.3 | 0.4 | <0.1 | <0.1 | 0.2 | 0.1 |
| 27 | 200.5 | 0.6 | 0.5 | 0.3 | 0.3 | 0.3 | <0.1 | <0.1 | 0.2 | 0.2 |
| 30 | 203.6 | 0.8 | 0.5 | 0.6 | 0.4 | 0.3 | <0.1 | <0.1 | 0.5 | 0.4 |
| 33 | 209.9 | 0.9 | 0.5 | 0.8 | 0.3 | 0.3 | 0.1 | <0.1 | 0.7 | 0.3 |
| 36 | 217.5 | 0.9 | 0.5 | 0.7 | 0.3 | 0.3 | <0.1 | <0.1 | 0.7 | 0.2 |
| 39 | 226.0 | 1.1 | 0.6 | 0.9 | 0.4 | 0.4 | 0.1 | <0.1 | 0.9 | 0.1 |
| 42 | 238.0 | 1.0 | 0.5 | 0.7 | 0.3 | 0.4 | <0.1 | <0.1 | 0.7 | 0.1 |
| 45 | 244.3 | 0.9 | 0.6 | 0.7 | 0.4 | 0.5 | <0.1 | <0.1 | 0.6 | 0.3 |
| 48 | 254.7 | 0.9 | 0.6 | 0.6 | 0.4 | 0.4 | <0.1 | <0.1 | 0.5 | 0.4 |
| 52 | 258.7 | 1.2 | 0.7 | 0.8 | 0.6 | 0.4 | <0.1 | <0.1 | 0.5 | 0.6 |
| 56 | 258.7 | 1.3 | 0.8 | 0.9 | 0.7 | 0.4 | <0.1 | <0.1 | 0.6 | 0.6 |
| 60 | 257.9 | 1.3 | 0.8 | 1.0 | 0.7 | 0.4 | <0.1 | <0.1 | 0.8 | 0.6 |
| 64 | 249.7 | 1.6 | 0.9 | 1.3 | 0.8 | 0.4 | <0.1 | 0.1 | 1.0 | 0.8 |
| 68 | 233.1 | 1.6 | 1.2 | 1.1 | 1.1 | 0.4 | <0.1 | 0.1 | 0.9 | 0.7 |



**Table A8.** Temperature error budget for daytime tropics. All uncertainties are $1\sigma$.

| Altitude | Temp. | Total Error | Random Error | Syst. Error | Meas. Noise | Gain Calibr. | Spectral Shift | $CO_2$- VMR | Spectrosc. Data | Instrument Line Shape |
|---|---|---|---|---|---|---|---|---|---|---|
| (km) | (K) | (K) | (K) | (K) | (K) | (K) | (K) | (K) | (K) | (K) |
| 15 | 200.4 | 1.6 | 0.7 | 1.4 | 0.6 | 0.2 | <0.1 | <0.1 | 1.2 | 0.8 |
| 18 | 197.6 | 0.9 | 0.4 | 0.8 | 0.4 | 0.1 | <0.1 | <0.1 | 0.8 | <0.1 |
| 21 | 210.7 | 0.8 | 0.4 | 0.7 | 0.2 | 0.2 | <0.1 | <0.1 | 0.7 | 0.1 |
| 24 | 214.7 | 0.6 | 0.4 | 0.4 | 0.3 | 0.3 | <0.1 | <0.1 | 0.3 | 0.1 |
| 27 | 221.1 | 0.8 | 0.4 | 0.7 | 0.3 | 0.3 | <0.1 | <0.1 | 0.7 | 0.1 |
| 30 | 228.7 | 0.9 | 0.4 | 0.8 | 0.3 | 0.3 | <0.1 | <0.1 | 0.6 | 0.4 |
| 33 | 235.5 | 0.9 | 0.4 | 0.7 | 0.3 | 0.3 | 0.1 | <0.1 | 0.6 | 0.4 |
| 36 | 241.9 | 0.8 | 0.5 | 0.6 | 0.3 | 0.3 | 0.1 | <0.1 | 0.4 | 0.4 |
| 39 | 247.3 | 0.7 | 0.5 | 0.5 | 0.3 | 0.4 | 0.1 | <0.1 | 0.4 | 0.2 |
| 42 | 252.8 | 0.7 | 0.6 | 0.4 | 0.3 | 0.5 | 0.1 | <0.1 | 0.4 | 0.1 |
| 45 | 256.7 | 0.6 | 0.6 | 0.3 | 0.3 | 0.5 | <0.1 | <0.1 | 0.2 | 0.2 |
| 48 | 256.7 | 0.7 | 0.6 | 0.4 | 0.4 | 0.4 | <0.1 | <0.1 | 0.3 | 0.3 |
| 52 | 255.9 | 1.0 | 0.6 | 0.8 | 0.5 | 0.3 | <0.1 | <0.1 | 0.5 | 0.6 |
| 56 | 248.5 | 1.3 | 0.7 | 1.1 | 0.5 | 0.4 | <0.1 | <0.1 | 0.8 | 0.8 |
| 60 | 236.6 | 1.5 | 0.8 | 1.3 | 0.7 | 0.5 | <0.1 | <0.1 | 1.0 | 0.8 |
| 64 | 221.1 | 1.6 | 1.0 | 1.3 | 0.8 | 0.5 | <0.1 | 0.1 | 1.0 | 0.8 |
| 68 | 209.0 | 1.5 | 1.2 | 0.9 | 1.1 | 0.5 | <0.1 | 0.1 | 0.6 | 0.6 |



**Table A9.** Temperature error budget for nighttime tropics. All uncertainties are $1\sigma$.

| Altitude | Temp. | Total Error | Random Error | Syst. Error | Meas. Noise | Gain Calibr. | Spectral Shift | $CO_2$-VMR | Spectrosc. Data | Instrument Line Shape |
|---|---|---|---|---|---|---|---|---|---|---|
| (km) | (K) | (K) | (K) | (K) | (K) | (K) | (K) | (K) | (K) | (K) |
| 12 | 225.3 | 2.5 | 0.9 | 2.3 | 0.7 | 0.5 | <0.1 | <0.1 | 2.0 | 1.2 |
| 15 | 202.7 | 2.1 | 0.7 | 2.0 | 0.6 | 0.2 | <0.1 | <0.1 | 1.7 | 1.1 |
| 18 | 199.8 | 0.6 | 0.4 | 0.4 | 0.3 | 0.1 | <0.1 | <0.1 | 0.4 | 0.1 |
| 21 | 208.4 | 0.8 | 0.4 | 0.6 | 0.3 | 0.2 | <0.1 | <0.1 | 0.6 | 0.1 |
| 24 | 214.8 | 0.8 | 0.4 | 0.7 | 0.3 | 0.3 | <0.1 | <0.1 | 0.6 | 0.1 |
| 27 | 222.7 | 0.8 | 0.4 | 0.7 | 0.3 | 0.3 | <0.1 | <0.1 | 0.7 | 0.1 |
| 30 | 231.1 | 0.9 | 0.4 | 0.8 | 0.3 | 0.3 | <0.1 | <0.1 | 0.6 | 0.4 |
| 33 | 236.5 | 0.8 | 0.4 | 0.6 | 0.3 | 0.3 | 0.1 | <0.1 | 0.4 | 0.5 |
| 36 | 241.3 | 0.8 | 0.5 | 0.6 | 0.3 | 0.4 | 0.1 | <0.1 | 0.4 | 0.3 |
| 39 | 246.8 | 0.8 | 0.5 | 0.6 | 0.3 | 0.4 | 0.1 | <0.1 | 0.5 | 0.2 |
| 42 | 254.1 | 0.8 | 0.6 | 0.5 | 0.3 | 0.5 | 0.1 | <0.1 | 0.5 | 0.1 |
| 45 | 259.1 | 0.7 | 0.6 | 0.4 | 0.3 | 0.5 | <0.1 | <0.1 | 0.4 | 0.2 |
| 48 | 258.5 | 0.8 | 0.6 | 0.6 | 0.4 | 0.4 | <0.1 | <0.1 | 0.5 | 0.3 |
| 52 | 256.5 | 0.9 | 0.6 | 0.6 | 0.5 | 0.4 | <0.1 | <0.1 | 0.4 | 0.5 |
| 56 | 253.0 | 1.2 | 0.7 | 0.9 | 0.5 | 0.4 | <0.1 | <0.1 | 0.7 | 0.6 |
| 60 | 245.9 | 1.4 | 0.8 | 1.1 | 0.6 | 0.5 | <0.1 | <0.1 | 0.9 | 0.8 |
| 64 | 230.1 | 1.8 | 1.0 | 1.4 | 0.8 | 0.6 | <0.1 | 0.1 | 1.2 | 0.8 |
| 68 | 211.0 | 1.7 | 1.2 | 1.1 | 1.1 | 0.6 | <0.1 | 0.1 | 1.1 | 0.3 |

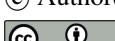



**Table B1.** Sample MIPAS temperature noise error correlation values between adjacent retrieval altitudes for selected altitudes of a limb scan on 12 July, 2009. $c_{n,n+1}$, $c_{n,n+2}$,... mark the first, second, ... off-diagonal elements of the respective $n^{\text{th}}$ row of the noise error correlation matrix.

| Altitude Index $n$ | Altitude (km) | $c_{n,n+1}$ | $c_{n,n+2}$ | $c_{n,n+3}$ | $c_{n,n+4}$ | $c_{n,n+5}$ | $c_{n,n+6}$ | $c_{n,n+7}$ |
|---|---|---|---|---|---|---|---|---|
| 7 | 9 | 8.56e-02 | -1.38e-01 | -3.99e-02 | 7.00e-02 | 6.26e-02 | 2.64e-02 | 1.61e-02 |
| 10 | 12 | -2.60e-02 | -1.70e-01 | -5.00e-02 | 3.40e-02 | 2.91e-02 | 4.95e-03 | -1.17e-03 |
| 13 | 15 | 1.02e-01 | -5.38e-02 | 2.74e-02 | 8.53e-02 | 4.61e-02 | 1.20e-02 | 1.02e-02 |
| 16 | 18 | -2.05e-02 | -1.15e-01 | 1.25e-02 | 7.50e-02 | 3.48e-02 | 7.88e-03 | -7.61e-04 |
| 19 | 21 | -6.65e-02 | -1.97e-01 | -1.25e-02 | 2.48e-02 | 3.24e-02 | 1.03e-02 | -1.93e-03 |
| 22 | 24 | -1.93e-02 | -1.71e-01 | -1.29e-01 | 1.73e-02 | 3.38e-02 | 3.25e-02 | 7.96e-03 |
| 25 | 27 | 1.52e-01 | -1.47e-01 | -1.15e-01 | -7.66e-02 | 1.04e-02 | 1.27e-02 | 7.72e-03 |
| 28 | 30 | 1.14e-02 | -2.66e-01 | -2.13e-01 | -9.70e-02 | -7.31e-03 | 3.43e-02 | 3.72e-02 |
| 31 | 33 | 2.34e-01 | -7.49e-02 | -1.52e-01 | -1.03e-01 | -6.00e-03 | 4.09e-02 | 4.96e-02 |
| 34 | 36 | 1.93e-01 | -9.54e-02 | -1.68e-01 | -1.09e-01 | -2.66e-02 | 1.27e-02 | 2.62e-02 |
| 37 | 39 | 1.67e-01 | -8.35e-02 | -1.37e-01 | -8.65e-02 | -1.46e-02 | 2.11e-02 | 3.15e-02 |
| 40 | 42 | 2.21e-01 | -5.31e-02 | -1.32e-01 | -8.59e-02 | -1.53e-02 | 2.50e-02 | 3.96e-02 |
| 43 | 45 | 2.29e-01 | -8.10e-03 | -1.26e-01 | -1.28e-01 | -9.20e-02 | 1.76e-02 | 1.27e-02 |
| 46 | 48 | 1.91e-01 | -1.74e-02 | -8.55e-02 | -4.06e-02 | -2.20e-02 | -1.05e-02 | -1.17e-02 |
| 49 | 52 | -1.94e-01 | -7.32e-02 | -1.60e-02 | -2.36e-02 | -5.47e-03 | 1.34e-02 | 1.54e-02 |
| 51 | 56 | -4.51e-04 | -1.27e-01 | -2.47e-03 | 9.91e-02 | 8.76e-02 | 6.66e-02 | 4.50e-02 |
| 53 | 60 | -5.78e-04 | -6.78e-02 | -2.48e-02 | 2.34e-03 | 1.72e-02 | 2.83e-02 | 4.02e-02 |
| 55 | 64 | 3.69e-01 | 7.87e-02 | -1.74e-02 | 1.82e-02 | 7.10e-02 | 1.06e-01 | 1.31e-01 |
| 57 | 68 | 3.29e-01 | -1.25e-01 | -2.07e-01 | -1.95e-01 | -1.47e-01 | -2.84e-02 | 1.40e-02 |



*Author contributions.*  Since the results presented in this paper are a team effort, the following list of author contributions is by no means
comprehensive. Instead only the most prominent contributions are listed. MK developed the retrieval setup, coordinated, partly performed
related test calculations, contributed graphics, and had the final editorial responsibility for this paper. TvC wrote large parts of the text,
organized related discussions and cared about TUNER compliance of error estimates. BF provided the parameterized NLTE approach and
implemented the updated frequency calibration and offset correction. BF, MGC and MLP took care that the retrieval setup was developed
in a way that inter-consistence with the retrieval setups of middle and upper atmospheric measurement modes was maintained. Furthermore
they provided the $CO_2$ uncertainties. MLP and BF built the a priori temperature distributions from the WACCM data. NG was responsible
for spectroscopy issues, error estimation calculations, carried out some of the retrieval tests, and contributed graphics. UG provided and
maintained the retrieval software including data archive. He further has been responsible for level-1b data import and quality control. SK
and A. Linden ran the retrievals and visualized the results. SK contributed graphics. AK was responsible for L1-related issues. A. Laeng
contributed to quality control. DM provided the WACCM calculations. GPS identified issues to be solved and took care of quality control.
All authors suggested solutions for various problems encountered during the development phase and critically discussed the results as well
as the manuscript.

*Competing interests.*  The authors declare that they have no conflict of interest.



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
