# Peer review of "IMK/IAA MIPAS temperature retrieval version 8: nominal measurements"

_Atmospheric Measurement Techniques, 2020_

## Referee Comment (RC1) · Chris Boone (Referee) · 21 Jan 2021

This article describes a new temperature retrieval from Michelson Interferometer for Passive Atmospheric Sounding (MIPAS) measurements, employing an updated set of radiance spectra along with upgrades to the processing scheme compared to previous processing versions. The paper is well-written, well-organized, clear, and comprehensive. I just have a few minor comments.

I would have liked to see a few more words on how pressure is handled. The comment on page 8 ["Between 43 and 53 km, a smooth transition between ECMWF and bias-corrected WACCM temperatures is obtained by linear interpolation along with hydrostatic correction of pressures at the given geometric altitudes."] seems to imply that

pressure is fixed to a priori information in the analysis and makes me wonder if perhaps the source of pressure information is different below 43 km versus above 53 km. The significance of this question comes from the following comment:

Page 24, line 553: V8 engineering tangent altitudes are, on average, lower than the retrieved ones by about 200 m below 40 km and by about 50 m above

I can imagine that the onset of refraction effects below 40 km might contribute to larger discrepancies in this altitude region, but I would perhaps naively expect such errors to increase with decreasing altitude. As described here, the errors seem to be more of a step function, possibly suggesting something in the analysis that generates a $\sim$150 m discrepancy below 40 versus above 40 km. Obvious candidates would be pressure (from page 8, the described hydrostatic correction of pressures between 43 and 53 km), or from Table 2, there is a set of three microwindows containing strong $CO_2$ lines where the lower altitude limit is 42 km. Discrepancies between these microwindows and other microwindows used in the analysis below 42 km could give rise to an apparent step in pointing. The latter possibility could be tested by adjusting the lower altitude limit of the three microwindows in question from 42 km up to 55 or 60 km and see if the discrepancy from engineering information changes for the region just above 40 km.

It would be good to say a few words on the determination of instrument pointing (tangent heights). I assume it basically amounts to ensuring hydrostatic equilibrium is maintained for the combination of assumed pressure profile and retrieved temperature profile, but it is not clear from the text.

The apparent step function in the discrepancy from engineering information below 40 km versus above 40 km is not reflected in the error estimates. If the source of the step function is a problem in the assumed pressure profile, the retrieved tangent height will mostly compensate, but there might be a 'second order' contribution to the retrieved temperature error, along with a 150 m altitude registration offset in the two altitude regions.

Even if there is a 150 m step, though, things are still better off than the previous processing version. I am not advocating further investigations at this time or significant changes to the paper. Just a few words on how pressure is handled and a brief description of the nature of the tangent height determination (e.g., ensuring hydrostatic equilibrium) would suffice.

>Page 10, line 248: The cause of the continuum signal from high altitudes is presumably meteoric dust

I do not know what magnitude of continuum levels are being discussed, but it would surprise me if that were the case. I can see there being measureable scattering effects for lidars operating in the visible, but we are talking about the thermal infrared here. Contributions to the spectrum from background sulfate aerosols could extend up to about 40 km. However, my first inclination would be far wing contributions from the nearby strong $CO_2$ Q-branch that are missing in the calculation (assuming your calculation does not extend that far in wavenumber), or the shape of the far wing contribution isn't quite right (e.g., it should be sub-Lorentzian or line mixing contributions are missing or not quite right). The shape (as a function of wavenumber) for the retrieved continuum parameters might give a clue. If the far wing lines are missing in the calculation for a set of microwindows, if the apparent continuum is larger for microwindows closer to the $CO_2$ Q-branch, that could suggest far wing effects are the source.

Note that I cannot say with certainty that the continuum is not associated with meteoric dust. Perhaps it is, but it would surprise me.

>Page 10, line 266: the offset can vary independently between microwindows

The sources of offsets that I can imagine would all at least vary smoothly with wavenumber. Self-emission of the instrument, which would provide an offset with a blackbody curve appropriate to the instrument temperature. Deficiencies in the detector non-linearity correction. Hard to say what the shape might be, but would it be random? Channeling artifacts, which would have a sinusoidal variation with wavenumber (or a superposition of sine waves if there are multiple contributions), but if your microwindows sample the sine pattern (or the beating pattern from multiple overlapping sine waves) at various locations, I suppose it might look vaguely random.

Is there typically a lot a scatter as a function of wavenumber in the retrieved offset values? Is there a physical explanation attributed to the offsets?

>Page 17: Random errors are errors which explain the standard deviation of the differences between measurements of the same state variable by two different instruments

I know what you are saying, but the wording implies a narrower definition of random error than I would like. How about the following:

When comparing measurements of the same state variable by two different instruments, random errors are errors that contribute to the intrinsic variability (standard deviation) of the differences.

——————— Minor edits and wording suggestions: ——————

>Page 3, line 56: exemplary results

'Exemplary' means perfect, flawless, the best of its kind. While it is possible you were suggesting your results are perfect, I wonder if you meant 'example results.'

>Page 4, line 91: MIPAS spectra are analyzed with constrained nonlinear least squares fit

. . .with a constrained. . .

>Page 4, line 110: but are least interfered of gases of unknown abundancy

Awkward phrasing. Suggest something like 'with minimal contributions from gases of unknown abundancy'

>Page 6, line 154: for the RR measurements)

delete the ')'

[Figure]

>Page 6, line 163: following implementation of the altitude dependence

. . .the following implementation. . .

>Page 20, line 480: and we investigate, to which degree MIPAS provides

delete the ','

>Page 21, line 500: it is obvious, that the differences

delete the ','

>Page 24, line 534: The example shown in the lower panel demonstrates, that MIPAS is

delete the ','
* * *

---

## Referee Comment (RC2) · Anonymous Referee #2 · 10 Feb 2021

General comments:

This is a very well and clearly written paper on the most recent MIPAS temperature data product retrieved with the IMK/IAA processor from the MIPAS nominal measurements. The paper fits well into AMT and the MIPAS special issue and I don't have any major objections against the publication of the manuscript. In my opinion minor revisions are required for the paper to become acceptable for publication and I ask the authors to consider the specific comments below.

Specific comments:

Line 44: ".. degraded spectral resolution reduced resolution"

Perhaps "reduced resolution" can be italicized of put in quotation marks to make it

easier to read. I had to read the sentence, particularly "degraded spectral resolution reduced resolution" several times.

Line 71: "correction led to too small values"

Unclear, what "values" refers to here: correction values, radiances or temperatures (probably radiances)? Please clarify.

Line 137: "and then fitting a linear regression function to the shift values, which are calculated for the single microwindows."

It would be good to mention how well the frequency shift values for the different microwindows can be approximated by a straight line.

Line 154: "measurements)" Closing parenthesis can be deleted.

Line 182: "occuring" -> "occurring"

Line 196: "is obtained by linear interpolation along with hydrostatic correction of pressures at the given geometric altitudes."

I don't really understand what was done here. Can you rephrase or add an additional sentence?

Line 203: "Since limb measurements used for one profile retrieval cover, depending on the measurement mode, about 1600 to 2200 km in the horizontal,"

I wonder, why this distance is so large. What does it refer to exactly? What is duration of a limb scan?

Section 3.5: Horizonal variability

The approach you used to consider horizontal variability seems very good. I suggest mentioning the horizontal resolution of your model atmosphere. This is not mentioned, as far as I can tell.

Section 3.7: Is the background continuum spectrally neutral? Probably yes, but it

should perhaps be mentioned explicitly.

Line 248: "The cause of the continuum signal from high altitudes is presumably meteoric dust"

Just out of interest: Is there any chance the measurements can be used to identify meteoric dust? Or has this been attempted already?

Section 3.8: Does the offset have a constant value for all wavenumbers of a microwindow?

Section 3.11: I suggest mentioning which process/reaction leads to vibrational populations being removed from LTE. If it's several processes, perhaps the most important one can be mentioned.

Line 310: "are use" -> "are used"

Line 328: "The atmospheric conditions under consideration are northern and southern polar winter, polar summer.. "

This is only a minor issue, but does polar summer include both hemispheres? I tried to count, whether it is nine scenarios and was a bit confused.

Section 4: I Suggest mentioning how the individual error sources were added to determine the total error.

Figure 3: Please explain the meaning of the red crosses and plus signs at the bottom of the figure.

Line 366: Please explain or spell out "IF16"

Line 371: "Section3.2" -> "Section 3.2"

Line 461: I suggest replacing "cold temperatures" by "low temperatures" because temperature cannot be cold, strictly speaking.

Line 487 – 498: It would be good to provide more quantitative information here. How

large was the drift before, compared to other sources (which sources) and how large is it now?

Figure 8: Suggest to mention the years in the caption of this figure, too.

Figure 9 and related discussion: One can see the differences, but one doesn't know which product agrees better with the true T-field. The discussions of the differences between versions 8 and 5 should be complemented by more quantitative comparisons with independent measurements. Perhaps you can simply refer to existing validation studies for V5.

Line 556: "The standard deviations .. was" -> "The standard deviations .. were"

Appendix A: The tables A1 to A9 differ in the altitude range shown. I guess this was done on purpose? If yes, it would be good to mention it and mention the reasons for the different altitude ranges.

---

## Author Comment (AC2) · 9 Mar 2021

**Replies to Anonymous Referee #2**

We thank the referee for his corrections and suggestions which we much appreciate, and we are confident that their implementation is beneficial for the readability and quality of our manuscript.

Questions/comments of the referee are marked by **RC:** and set in *slanted font*.

The suggested corrections of lines 154, 182, 310, 371, 461, 556 have been imple-

mented.

**RC:** *Line 44: ".. degraded spectral resolution reduced resolution" Perhaps "reduced resolution" can be italicized of put in quotation marks to make it easier to read. I had to read the sentence, particularly "degraded spectral resolution reduced resolution" several times.*

**Reply:** We changed the sentence to: For this second operation phase with degraded spectral resolution we shall use the designation "reduced spectral resolution" (RR) period.

**RC:** *Line 71: "correction led to too small values" Unclear, what "values" refers to here: correction values, radiances or temperatures (probably radiances)? Please clarify.*

**Reply:** It is the temperature values, which is stated in the text now. In the cited report of Hubert et al. temperature (among other quantities) from several MIPAS data versions is compared with correlative measurements.

**RC:** *Line 137: "and then fitting a linear regression function to the shift values, which are calculated for the single microwindows." It would be good to mention how well the frequency shift values for the different microwindows can be approximated by a straight line.*

**Reply:** Text with appropriate numbers has been added after the cited sentence.

**RC:** *Line 196: "is obtained by linear interpolation along with hydrostatic correction of*

*pressures at the given geometric altitudes." I don't really understand what was done here. Can you rephrase or add an additional sentence?*

**Reply:** The text has been expanded with a description of the temperature calculation in the transition region and of the pressure integration.

**RC:** *Line 203: "Since limb measurements used for one profile retrieval cover, depending on the measurement mode, about 1600 to 2200 km in the horizontal," I wonder, why this distance is so large. What does it refer to exactly? What is duration of a limb scan?*

**Reply:** The sentence has been split in two and partly rewritten. We now mention the horizontal ranges 260 to 440 km from which 95% of the radiance information comes from, and include a reference for these values. The numbers in the original version were simply the length of the line of sight between the top of the atmosphere (assumed to be about 100 km) and the tangent point, multiplied by two because there is a path segment behind the tangent point and another one in front of the tangent point

**RC:** *Section 3.5: Horizontal variability The approach you used to consider horizontal variability seems very good. I suggest mentioning the horizontal resolution of your model atmosphere. This is not mentioned, as far as I can tell.*

**Reply:** The horizontal resolution of the used ECMWF ERA-Interim data set is given now in the text.

**RC:** *Section 3.7: Is the background continuum spectrally neutral? Probably yes, but it should perhaps be mentioned explicitly.*

[Figure]

**Reply:** It is not clear to us what "spectrally neutral" means. The background continuum is wavenumber-independent within a microwindow in terms of the absorption cross-section, which implies, via the Planck function, a slight wavenumber-dependence within the microwindow.

**RC:** *Line 248: "The cause of the continuum signal from high altitudes is presumably meteoric dust" Just out of interest: Is there any chance the measurements can be used to identify meteoric dust? Or has this been attempted already?*

**Reply:** The only information we have is the paper by Neely II et al (2011), cited in the paper. We too are curious whether this hypothesis can be independently corroborated. To make more clear that we consider it a hypothesis, we changed "presumably" to "possibly" in the text.

**RC:** *Section 3.8: Does the offset have a constant value for all wavenumbers of a microwindow?*

**Reply:** Yes. The text has slightly been changed to make this more clear.

**RC:** *Section 3.11: I suggest mentioning which process/reaction leads to vibrational populations being removed from LTE. If it's several processes, perhaps the most important one can be mentioned.*

**Reply:** A brief description of the cause of the population of the NLTE states at 15 $\mu$m has been added to the text.

**RC:** *Line 328: "The atmospheric conditions under consideration are northern and southern polar winter, polar summer.. " This is only a minor issue, but does polar summer include both hemispheres? I tried to count, whether it is nine scenarios and was a bit confused.*

**Reply:** We meanwhile have error estimates for a larger number of atmospheric conditions available (now also distinguishing N and S polar summer, and many more). We will adjust the text and the presentation of data accordingly.

**RC:** *Section 4: I Suggest mentioning how the individual error sources were added to determine the total error.*

**Reply:** Since we consider all error sources as independent from each other they are added quadratically to give the total error. This is explained now in Section 4.1.

**RC:** *Figure 3: Please explain the meaning of the red crosses and plus signs at the bottom of the figure.*

**Reply:** These symbols indicate the position of the measurement and give the lighting conditions (cross: night, plus: day). This is stated in the caption now.

**RC:** *Line 366: Please explain or spell out "IF16"*

**Reply:** The term "IF16" has been cancelled. Instead we have added some details about what kind of data is used and why (only) these data could be used for this

specific method to determine gain uncertainties.

**RC:** *Line 487–498: It would be good to provide more quantitative information here. How large was the drift before, compared to other sources (which sources) and how large is it now?*

**Reply:** Section 5.2.1 has been completely rewritten and the figure has been exchanged. We now present drifts in temperature of MIPAS V5 wrt two reference instruments, as extracted from a paper by McLandress et al. (2015), and then use the accordingly calculated drift between MIPAS V5 and V8 temperatures to infer that there indeed has been an improvement in the data quality.

**RC:** *Figure 8: Suggest to mention the years in the caption of this figure, too*

**Reply:** OK, done.

**RC:** *Figure 9 and related discussion: One can see the differences, but one doesn't know which product agrees better with the true T-field. The discussions of the differences between versions 8 and 5 should be complemented by more quantitative comparisons with independent measurements. Perhaps you can simply refer to existing validation studies for V5.*

**Reply:** Since this deficiency in V5 occurs only in specific situations, it was not detected by temperature validation studies of nominal mode data. However, we now use the V5 middle atmosphere data (which has been validated by García-Comas et al., 2014) as a reference to show that V8 nominal mode data indeed has improved over V5 nominal

data. Figure 9 and caption, as well as the text of Section 5.1.3, have been extended accordingly.

**RC:** *Appendix A: The tables A1 to A9 differ in the altitude range shown. I guess this was done on purpose? If yes, it would be good to mention it and mention the reasons for the different altitude ranges.*

**Reply:** There are two reasons: First the lowest altitude of MIPAS nominal mode observations varies along the orbit/latitude. However, the second, and more important, factor is that the errors are only defined at altitudes where the spectra are not contaminated by IR-emission of clouds. The cloud altitude strongly depends on latitude and season. This is now explained in the text of Appendix A.
* * *

---

## Author Response (AR1)

We thank the referees for their corrections and suggestions which we much appreciate, and we are confident that their implementation is beneficial for the readability and quality of our manuscript.

Questions/comments of the referees are marked by **RC:** and set in *slanted font*.

The line numbers given under the items marked **Text Change** refer to the PDF-document which shows highlighted differences between the revised and the original version of our manuscript.

**Replies to Referee #1**

All minor edits and wording suggestions of the referee have been implemented.

**RC:** *I would have liked to see a few more words on how pressure is handled. The comment on page 8 ["Between 43 and 53 km, a smooth transition between ECMWF and bias-corrected WACCM temperatures is obtained by linear interpolation along with hydrostatic correction of pressures at the given geometric altitudes."] seems to imply that pressure is fixed to a priori information in the analysis and makes me wonder if perhaps the source of pressure information is different below 43 km versus above 53 km. The significance of this question comes from the following comment: Page 24, line 553: V8 engineering tangent altitudes are, on average, lower than the retrieved ones by about 200 m below 40 km and by about 50 m above*

**Reply:** The method which leads to the a priori temperature and pressure values above 43 km is now described in more detail in the text. Only the pressure at 20 km altitude (or at the lowest tangent altitude above 20 km with valid spectral information) is taken from the apriori profile for the calculation of the pressure profile during each iteration step, making use of the hydostatic equation. This is described in some detail in the 2003 paper by von Clarmann et al., cited in the text. Sentences that shortly state this and which contain a citation of the paper where details of the method can be found have been added to the text of Secs. 3, 3.1, and 3.4. From this it should be clear that one would not expect a visible influence of the transition region pressure on the line-of-sight retrieval. We realize that the referee's suspicion that there might be an influence probably is based upon our maybe too sloppy description of the course with altitude of the differences between engineering tangent altitudes and retrieved tangent altitudes under item 3 of Sec. 5.4. The keywords here are "on average". Actually it is not true that there is a jump at 40 km in the differences, but only in the averages from 0-40 km and 40-70 km. However, we unfortunately had to realize that our presentation in the manuscript was erroneously based on FR data alone. Hence, the text has been reworked to also take into account the RR period. We try to be more specific about the altitude course of the differences of quantities to avoid the (wrong) impression that

there are steps or jumps.

**Text Change:** Text has been changed in Sections 3 (lines 88–90 were added), 3.1 (lines 110–113 were added), and 3.4. (lines 205–211 were added) to clarify the method how pressure is calculated. Section 5.4. has been partly rewritten to include statements on FR and RR data (erroneously only FR before).
* * *
**RC:** *I can imagine that the onset of refraction effects below 40 km might contribute to larger discrepancies in this altitude region, but I would perhaps naively expect such errors to increase with decreasing altitude.*

**Reply:** Right. Actually these errors kick in below 20 km. This can be nicely seen in comparisons between the engineering tangent altitudes of V5 and V8, since in the L1b-processor refraction has been implemented after V5 processing. We briefly mention this in the reworked Section 5.4.

**Text Change:** Section 5.4. has been partly rewritten and now also includes a statement on the effect of refraction (item 1, line 601).
* * *
**RC:** *As described here, the errors seem to be more of a step function, possibly suggesting something in the analysis that generates a $< 150$ m discrepancy below 40 versus above 40 km. Obvious candidates would be pressure (from page 8, the described hydrostatic correction of pressures between 43 and 53 km), or from Table 2, there is a set of three microwindows containing strong CO2 lines where the lower altitude limit is 42 km. Discrepancies between these microwindows and other microwindows used in the analysis below 42 km could give rise to an apparent step in pointing. The latter possibility could be tested by adjusting the lower altitude limit of the three microwindows in question from 42 km up to 55 or 60 km and see if the discrepancy from engineering information changes for the region just above 40 km.*

**Reply:** As already mentioned above, there actually is no step function. There are no retrieved quantities which change (more or less) abruptly at an altitude of 40 km, neither at any other altitude. However, we much appreciate the Referee's comments, obviously motivated by the wish to help us to overcome an error, which, if it actually existed, would be an unpleasant one.

**Text Change:** Section 5.4., especially item 3., has been partly rewritten to make this point more clear.
* * *
**RC:** *It would be good to say a few words on the determination of instrument pointing (tangent heights). I assume it basically amounts to ensuring hydrostatic equilibrium is maintained for the combination of assumed pressure profile and retrieved temperature profile, but it is not clear from the text.*

**Reply:** We have added text to clarify this to Sections 3, 3.1, and 3.4. Since the method has already been described by von Clarmann et al. (2003) we kept these additional text items short and cite this work where appropriate.

**Text Change:** Text and the proper citation have been added (lines 88–90, 110–113, 126, and 205–211).
* * *
**RC:** *The apparent step function in the discrepancy from engineering information below 40 km versus above 40 km is not reflected in the error estimates. If the source of the step function is a problem in the assumed pressure profile, the retrieved tangent height will mostly compensate, but there might be a "second order" contribution to the retrieved temperature error, along with a 150 m altitude registration offset in the two altitude regions. Even if there is a 150 m step, though, things are still better off than the previous processing version. I am not advocating further investigations at this time or significant changes to the paper. Just a few words on how pressure is handled and a brief description of the nature of the tangent height determination (e.g., ensuring hydrostatic equilibrium) would suffice.*

**Reply:** The error estimates do not include the step because there is none.

**Text Change:** None.
* * *
**RC:** *Page 10, line 248: "The cause of the continuum signal from high altitudes is presumably meteoric dust." I do not know what magnitude of continuum levels are being discussed, but it would surprise me if that were the case. I can see there being measurable scattering effects for lidars operating in the visible, but we are talking about the thermal infrared here. Contributions to the spectrum from background sulfate aerosols could extend up to about 40 km. However, my first inclination would be far wing contributions from the nearby strong $CO_2$ Q-branch that are missing in the calculation (assuming your calculation does not extend that far in wavenumber), or the shape of the far wing contribution isn't quite right (e.g., it should be sub-Lorentzian or line mixing contributions are missing or not quite right). The shape (as a function of wavenumber) for the retrieved continuum parameters might give a clue. If the far wing lines are missing in the calculation for a set of microwindows, if the apparent continuum is larger for microwindows closer to the $CO_2$ Q-branch, that could*

*suggest far wing effects are the source. Note that I cannot say with certainty that the continuum is not associated with meteoric dust. Perhaps it is, but it would surprise me.*

**Reply:** We would be surprised if pressure broadening and related effects played such a prominent role particularly at these high altitudes where Doppler broadening becomes the dominating mechanism. However, nothing in our retrieval scheme depends on this since we use the empirically fitted continuum. Our remark on the meteoric dust is not more than a speculative, tentative explanation. To make the speculative nature of our statement clearer, we replaced "presumably" with "possibly" in the text.

**Text Change:** Wording changed (line 264).
* * *
**RC:** *Page 10, line 266: the offset can vary independently between microwindows The sources of offsets that I can imagine would all at least vary smoothly with wavenumber. Self-emission of the instrument, which would provide an offset with a blackbody curve appropriate to the instrument temperature. Deficiencies in the detector non-linearity correction. Hard to say what the shape might be, but would it be random? Channeling artifacts, which would have a sinusoidal variation with wavenumber (or a superposition of sine waves if there are multiple contributions), but if your microwindows sample the sine pattern (or the beating pattern from multiple overlapping sine waves) at various locations, I suppose it might look vaguely random. Is there typically a lot a scatter as a function of wavenumber in the retrieved offset values? Is there a physical explanation attributed to the offsets?*

**Reply:** Since MIPAS spectra are calibrated, the offset accounts only for the residual differences after calibration. Self-emission of the instrument should first order be calibrated away. Since the calibration measurements themselves are susceptible to noise, we suspect that the offset may well have a random component. Since ideally everything that can cause an offset should be calibrated away, discussions on the sources of the residual offset must remain speculative. Thus we prefer not to discuss this issue at any depth.

**Text Change:** None.
* * *
**RC:** *Page 17: "Random errors are errors which explain the standard deviation of the differences between measurements of the same state variable by two different instruments." I know what you are saying, but the wording implies a narrower definition of random error than I would like. How about the following: When comparing measurements of the same state variable by two different instruments, random errors are errors that contribute to the intrinsic variability (standard deviation) of the differences.*

**Reply:** The text is changed accordingly.

**Text Change:** Text has been changed (lines 433–435).

**Replies to Anonymous Referee #2**

The suggested corrections of lines 154, 182, 310, 371, 461, 556 (these line numbers refer to original version of the manuscript) have been implemented.

**RC:** *Line 44: ".. degraded spectral resolution reduced resolution" Perhaps "reduced resolution" can be italicized of put in quotation marks to make it easier to read. I had to read the sentence, particularly "degraded spectral resolution reduced resolution" several times.*

**Reply:** We changed the sentence to: For this second operation phase with degraded spectral resolution we shall use the designation "reduced spectral resolution" (RR) period.

**Text Change:** Text has been changed at lines 43–45.
* * *
**RC:** *Line 71: "correction led to too small values" Unclear, what "values" refers to here: correction values, radiances or temperatures (probably radiances)? Please clarify.*

**Reply:** It is the temperature values, which is stated in the text now. In the cited report of Hubert et al. temperature (among other quantities) from several MIPAS data versions is compared with correlative measurements.

**Text Change:** Text has been changed at lines 72–73.
* * *
**RC:** *Line 137: "and then fitting a linear regression function to the shift values, which are calculated for the single microwindows." It would be good to mention how well the frequency shift values for the different microwindows can be approximated by a straight line.*

**Reply:** Text with appropriate numbers has been added after the cited sentence.

**Text Change:** Text has been added (lines 145–147).

**RC:** *Line 196: "is obtained by linear interpolation along with hydrostatic correction of pressures at the given geometric altitudes." I don't really understand what was done here. Can you rephrase or add an additional sentence?*

**Reply:** The text has been expanded with a description of the temperature calculation in the transition region and of the pressure integration.

**Text Change:** Text has been added (lines 203–211).
* * *
**RC:** *Line 203: "Since limb measurements used for one profile retrieval cover, depending on the measurement mode, about 1600 to 2200 km in the horizontal," I wonder, why this distance is so large. What does it refer to exactly? What is duration of a limb scan?*

**Reply:** The sentence has been split in two and partly rewritten. We now mention the horizontal ranges 260 to 440 km from which 95% of the radiance information comes from, and include a reference for these values. The numbers in the original version were simply the length of the line of sight between the top of the atmosphere (assumed to be about 100 km) and the tangent point, multiplied by two because there is a path segment behind the tangent point and another one in front of the tangent point

**Text Change:** Text has been changed and a reference has been added (lines 217–220).
* * *
**RC:** *Section 3.5: Horizontal variability The approach you used to consider horizontal variability seems very good. I suggest mentioning the horizontal resolution of your model atmosphere. This is not mentioned, as far as I can tell.*

**Reply:** The horizontal resolution of the used ECMWF ERA-Interim data set is given now in the text.

**Text Change:** Text has been added (line 226).
* * *
**RC:** *Section 3.7: Is the background continuum spectrally neutral? Probably yes, but it should perhaps be mentioned explicitly.*

**Reply:** It is not clear to us what "spectrally neutral" means. The background continuum is wavenumber-independent within a microwindow in terms of the absorption cross-section, which implies, via the Planck function, a slight wavenumber-dependence

within the microwindow.

**Text Change:** None.
* * *
**RC:** *Line 248: "The cause of the continuum signal from high altitudes is presumably meteoric dust" Just out of interest: Is there any chance the measurements can be used to identify meteoric dust? Or has this been attempted already?*

**Reply:** The only information we have is the paper by Neely II et al (2011), cited in the paper. We too are curious whether this hypothesis can be independently corroborated. To make more clear that we consider it a hypothesis, we changed "presumably" to "possibly" in the text.

**Text Change:** Wording changed (line 264).
* * *
**RC:** *Section 3.8: Does the offset have a constant value for all wavenumbers of a microwindow?*

**Reply:** Yes. The text has slightly been changed to make this more clear.

**Text Change:** Text has been added (line 287).
* * *
**RC:** *Section 3.11: I suggest mentioning which process/reaction leads to vibrational populations being removed from LTE. If it's several processes, perhaps the most important one can be mentioned.*

**Reply:** A brief description of the cause of the population of the NLTE states at 15 $\mu$m has been added to the text.

**Text Change:** Text has been added (lines 300–302).
* * *
**RC:** *Line 328: "The atmospheric conditions under consideration are northern and southern polar winter, polar summer.. " This is only a minor issue, but does polar summer include both hemispheres? I tried to count, whether it is nine scenarios and was a bit confused.*

**Reply:** We meanwhile have error estimates for a larger number of atmospheric conditions available (now also distinguishing N and S polar summer, and many more). We

have adjusted the text and the presentation of data accordingly.

**Text Change:** Text as well as related figures and tables have been changed. See the last Section of this document, where the changes are discussed in more detail.
* * *
**RC:** *Section 4: I Suggest mentioning how the individual error sources were added to determine the total error.*

**Reply:** Since we consider all error sources as independent from each other they are added quadratically to give the total error. This is explained now in Section 4.1.

**Text Change:** Text has been added (line 377).
* * *
**RC:** *Figure 3: Please explain the meaning of the red crosses and plus signs at the bottom of the figure.*

**Reply:** These symbols indicate the position of the measurement and give the lighting conditions (cross: night, plus: day). This is stated in the caption now.

**Text Change:** Text has been added to the caption of Fig. 3.
* * *
**RC:** *Line 366: Please explain or spell out "IF16"*

**Reply:** For the new error calculations we abandoned the method mentioned in the original manuscript. Now we use data from Table 3 of Kleinert et al. (2018). This is stated in the new text version.

**Text Change:** Text has been changed (389–392).
* * *
**RC:** *Line 487–498: It would be good to provide more quantitative information here. How large was the drift before, compared to other sources (which sources) and how large is it now?*

**Reply:** Section 5.2.1 has been completely rewritten and the figure has been exchanged. We now present drifts in temperature of MIPAS V5 wrt two reference instruments, as extracted from a paper by McLandress et al. (2015), and then use the accordingly calculated drift between MIPAS V5 and V8 temperatures to infer that there indeed has

been an improvement in the data quality.

**Text Change:** Text of Section 5.2.1 has been rewritten (lines 509–533) and the figure and figure caption have been exchanged.
* * *
**RC:** *Figure 8: Suggest to mention the years in the caption of this figure, too*

**Reply:** OK, done.

**Text Change:** Figure caption text of Figure 8 has been extended.
* * *
**RC:** *Figure 9 and related discussion: One can see the differences, but one doesn't know which product agrees better with the true T-field. The discussions of the differences between versions 8 and 5 should be complemented by more quantitative comparisons with independent measurements. Perhaps you can simply refer to existing validation studies for V5.*

**Reply:** Since this deficiency in V5 occurs only in specific situations, it was not detected by temperature validation studies of nominal mode data. However, we now use the V5 middle atmosphere data (which has been validated by García-Comas et al., 2014) as a reference to show that V8 nominal mode data indeed has improved over V5 nominal data. Figure 9 and caption, as well as the text of Section 5.1.3, have been extended accordingly.

**Text Change:** The text of Section 5.2.3 has been extended (lines 557–562), and Figure 9 and caption have been exchanged.
* * *
**RC:** *Appendix A: The tables A1 to A9 differ in the altitude range shown. I guess this was done on purpose? If yes, it would be good to mention it and mention the reasons for the different altitude ranges.*

**Reply:** There are two reasons: First the lowest altitude of MIPAS nominal mode observations varies along the orbit/latitude. However, the second, and more important, factor is that the errors are only defined at altitudes where the spectra are not contaminated by IR-emission of clouds. The cloud altitude strongly depends on latitude and season. This is now explained in the text of Appendix A.

**Text Change:** Text has been added (lines 660–669).

**Text and other changes to the manuscript which are not directly linked to referee comments**

**Error estimates**

A complete recalculation of the error contributions to the total estimated temperature error has been performed. The new error estimation is based on larger samples, covering a larger number of atmospheric conditions. In particular, the new error estimation considers, contrary to the original submission, also the high spectral resolution measurement mode (years 2002-2004) results. This led to several changes of the manuscript:

- The abstract now contains the value ranges and attribution to representative atmospheric types corresponding to the new error estimation data set.

- The text of Section 4 has been changed in the same sense.

- Figures 1 and 2 are exchanged to show the revised error estimates.

- Tables A1 through A9 are exchanged to list the values of the revised error estimates.

The complete error estimates for day and night of spring, summer, autumn, and winter conditions at polar and middle latitudes, as well as for the tropics, are added as a supplement to the paper.

**Temperature drift**

Section 5.2.1 has been largely reworked. The statement that the temperature drift has considerably improved between V5 and V8 data is now based on a comparison of V5 data to external instruments (presented in McLandress et al. (2015)) and of V5 to V8 data. This is now explained and Fig. 6 has been exchanged to show the result of the comparisons.

**Pointing differences**

Section 5.4 has been largely rewritten. In the original manuscript by accident only the conditions for the FR period have been taken into account. The new text refers to FR and RR data.